# LONGWRITER: UNLEASHING 10,000+ WORD GENERATION FROM LONG CONTEXT LLMS

**Yushi Bai**[1], **Jiajie Zhang**[1], **Xin Lv**[2], **Linzhi Zheng**[1], **Siqi Zhu**[1],
**Lei Hou**[1], **Yuxiao Dong**[1], **Jie Tang**[1†], **Juanzi Li**[1†]
[1]Tsinghua University    [2]Zhipu AI

## ABSTRACT

Current long context large language models (LLMs) can process inputs up to 100,000 tokens, yet struggle to generate outputs exceeding even a modest length of 2,000 words. Through controlled experiments, we find that the model's effective generation length is inherently bounded by the sample it has seen during supervised fine-tuning (SFT). In other words, their output limitation is due to the scarcity of long-output examples in existing SFT datasets. To address this, we introduce AgentWrite, an agent-based pipeline that decomposes ultra-long generation tasks into subtasks, enabling off-the-shelf LLMs to generate coherent outputs exceeding 20,000 words. Leveraging AgentWrite, we construct LongWriter-6k, a dataset containing 6,000 SFT data with output lengths ranging from 2k to 32k words. By incorporating this dataset into model training, we successfully scale the output length of existing models to over 10,000 words while maintaining output quality. We also develop LongBench-Write, a comprehensive benchmark for evaluating ultra-long generation capabilities. Our 9B parameter model, further improved through DPO, achieves state-of-the-art performance on this benchmark, surpassing even much larger proprietary models. In general, our work demonstrates that existing long context LLM already possesses the potential for a larger output window–all you need is data with extended output during model alignment to unlock this capability. Our code & models are at: https://github.com/THUDM/LongWriter.

## 1 INTRODUCTION

Recent advancements in long context large language models (LLMs) have led to the development of models with significantly expanded memory capacities, capable of processing history exceeding 100,000 tokens in length (Anthropic, 2024; Reid et al., 2024; GLM et al., 2024). However, despite their ability to handle extensive inputs, current long-context LLMs struggle to generate equally lengthy outputs. To explore this limitation, we probe the maximum output length of state-of-the-art long-context models with multiple queries that require responses of varying lengths, for instance, "*Write a 10000-word article on the history of the Roman Empire*" (more details of this test in Sec. 2). From the result in Figure 1, we find that all models consistently fail to produce outputs beyond 2,000 words in length. Meanwhile, analysis of user interaction logs from WildChat (Zhao et al., 2024) reveals that over 1% of user prompts explicitly request outputs exceeding this limit, highlighting a pressing need in current research to overcome this limitation.

As a pilot study, we first investigate the underlying cause of the generation length limits observed in current models (Sec. 2). Our study reveals a key insight: the constraint on output length is primarily rooted in the characteristics of the Supervised Fine-Tuning (SFT) datasets. Specifically, we find that **a model's maximum generation length is effectively capped by the upper limit of output lengths present in its SFT dataset**, despite its exposure to much longer sequences during the pre-training phase (Xiong et al., 2024; Fu et al., 2024). This finding explains the ubiquitous 2,000-word generation limit across current models, as existing SFT datasets rarely contain examples exceeding this length. Furthermore, as many datasets are distilled from state-of-the-art LLMs (Chiang et al., 2023; Ding et al., 2023), they also inherit the output length limitation from their source models.

---

†Corresponding authors

To address this limitation, we introduce AgentWrite, a novel agent-based pipeline designed to leverage off-the-shelf LLMs to automatically construct extended, coherent outputs (Sec. 3). AgentWrite operates in two stages: First, it crafts a detailed writing plan outlining the structure and target word count for each paragraph based on the user's input. Then, following this plan, it prompts the model to generate content for each paragraph in a sequential manner. Our experiments validate that AgentWrite can produce high-quality and coherence outputs of up to 20,000 words.

Building upon the AgentWrite pipeline, we leverage GPT-4o to generate 6,000 long-output SFT data, namely *LongWriter-6k*, and add these data to train existing models. Notably, *LongWriter-6k* successfully unlocks the model's ability to generate well-structured outputs exceeding 10,000 words in length (Sec. 4). To rigorously evaluate the effectiveness of our approach, we develop the LongBench-Write benchmark, which contains a diverse set of user writing instructions, with output length specifications ranging from 0-500 words, 500-2,000 words, 2,000-4,000 words, and beyond 4,000 words. Evaluation on LongBench-Write shows that our 9B size model achieves state-of-the-art performance, even compared to larger proprietary models. We further construct preference data and use DPO (Rafailov et al., 2024) to help the model better follow long writing instructions and generate higher quality written content, which has also been proven effective through experiments.

To summarize, our work makes the following novel contributions:

- **Analysis of Generation Length Limits**: We identify the primary factor limiting the output length of current (long-context) LLMs, which is the constraint on the output length in the SFT data.
- **AgentWrite**: To overcome this limitation, we propose AgentWrite, which uses a divide-and-conquer approach with off-the-shelf LLMs to automatically construct SFT data with ultra-long outputs. Using this method, we construct the *LongWriter-6k* dataset.
- **Scaling Output Window Size of Current LLMs**: We incorporate the *LongWriter-6k* dataset into our SFT data, successfully scaling the output window size of existing models to 10,000+ words without compromising output quality. We show that DPO further enhances the model's long-text writing capabilities.

## 2 FINDING THE CAUSE OF THE BOUNDED GENERATION LENGTH LIMIT

First, we construct the *LongWrite-Ruler* evaluation to probe the generation length limits of LLMs. Then, we explore the reasons for their bounded generation length: By altering the maximum output length of the data in the model's SFT stage, we find that the maximum output length of the trained models on the LongWrite-Ruler test shows a significant positive correlation with the maximum output length of the SFT data. Note that throughout this paper, output length is measured in words (or characters for Chinese text) rather than tokens, as tokenization methods can vary across different models.

**LongWrite-Ruler**. To probe the maximum output length an LLM can provide, we construct a lightweight test: We create 8 different instructions, four each in Chinese and English, and vary the output length requirement "$L$" in the instructions. For example, "*Write a L-word article on the history of the Roman Empire*". During testing, we use $L \in \{1000, 2000, 5000, 10000, 20000, 30000\}$, resulting in a total of 48 test prompts (detailed test cases in Appendix B).

**Probing**. We measure the maximum output length of 4 open-source models and 4 proprietary models (details of our evaluated model in Table 5) on LongWrite-Ruler. During inference, we set the temperature to 0.5. For proprietary models, we configure the `max_tokens` parameter for generation to the maximum output length supported by the respective model's API call. For open-source models, we set it to 32k. In the output, we verify that no models produce truncated output due to the `max_tokens` constraint, which could have underestimated their maximum output length. Meanwhile, we observe almost no cases of repetitive content generation, which might have led to an overestimation. The results are visualized in Figure 1: For each length requirement (`x`-axis), we plot the average output length (`y`-axis) of the model across the 8 corresponding instructions. We use log-scale for `x`-axis and `y`-axis. We can observe from the figure that the maximum output length of all models is around 2k words. The effective output window size of proprietary models generally cannot reach their maximum token generation length. Furthermore, due to an increasing number of refusal cases, the average output length even decreases as the required length increases beyond 10k.

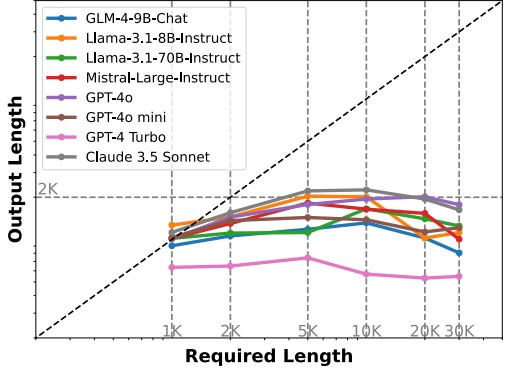

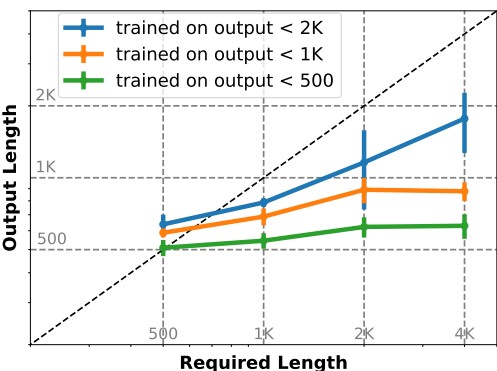

Figure 1: LongWriter-Ruler test demonstrates a maximum output length limitation of approximately 2k words for all models tested.

Figure 2: LongWriter-Ruler test of GLM-4-9B trained on SFT datasets of different maximum output lengths.

**Controlled experiment**. We hypothesize that the common 2,000-word output length limit is due to the inherent output length constraints present in SFT data, that is, "one can only speak as long as one has read". To test this hypothesis, we conduct a series of controlled experiments by altering the SFT data. In our experiments, we use GLM-4-9B (GLM et al., 2024) as the base model and select GLM-4's chat SFT data (a total of 180k data, which is a subset of GLM-4's entire SFT data) as the complete SFT dataset. To control the maximum output length of the SFT data, we filter out data with output lengths exceeding 500, 1,000, and 2,000 words, respectively. This results in three training sets, comprising 72%, 98%, and 99.9% of the original data, respectively.

We train GLM-4-9B model on these three training sets and measure the resulting models' maximum output length on LongWriter-Ruler (testing with $L \in \{500, 1000, 2000, 4000\}$). As shown in Figure 2, the model's maximum output length increases proportionally with the maximum output length in the SFT data, reaching approximately 600, 900, and 1,800 words, respectively. This increase in maximum output length also corresponds to an improvement in the model's average output length for instructions at each required length. This finding indicates that the model's output limit is due to insufficient output length in the SFT data. Moreover, this limitation cannot be overcome by LLM synthesized training data (Tunstall et al., 2023; Abdin et al., 2024) or through iterative SFT (Chen et al., 2024b; Burns et al., 2023), since data generated by existing models still cannot break through the length limit. In the following sections, we will explore the construction of SFT data with extended output lengths to further unleash the model's potential for longer output generation.

## 3 AGENTWRITE: AUTOMATIC DATA CONSTRUCTION

To utilize off-the-shelf LLMs for automatically generating SFT data with longer outputs, we design AgentWrite, a divide-and-conquer style agent pipeline (illustrated in Figure 3). AgentWrite first breaks down long writing tasks into multiple subtasks, with each subtask requiring the model to write only one paragraph. The model then executes these subtasks sequentially, and we concatenate the subtask outputs to obtain the final long output. Such an approach of breaking down a complex task into multiple subtasks using LLM agents has already been applied in various fields, such as problem-solving (Wu et al., 2023), software development (Qian et al., 2023), and model evaluation (Saha et al., 2024). Our work is the first to explore integrating planning to enable models to complete complex long-form writing tasks. We will introduce each step of AgentWrite in detail.

### 3.1 AGENTWRITE PIPELINE

**Step I: Plan**. Inspired by human writer's thought process, where a writer usually starts with making an overall plan for long writing tasks, typically involving outlining the structure and planning the content and length of each section. We utilize the planning capabilities of LLMs to output such a writing outline given a writing instruction, which includes the main content and word count requirements for each paragraph. Our prompt is presented in Appendix C.

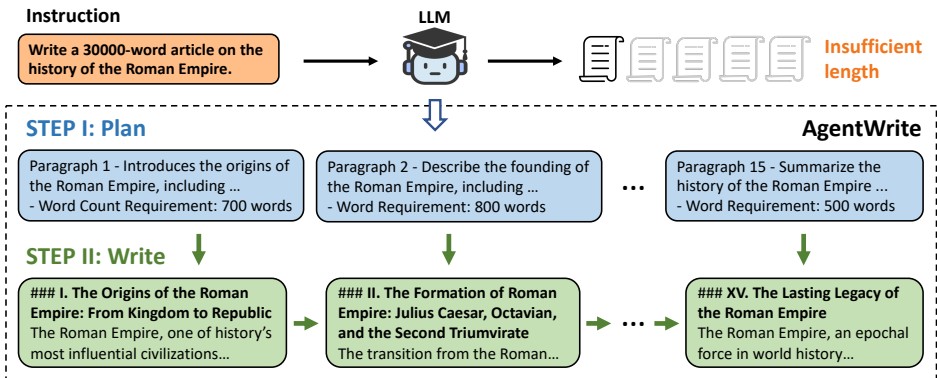

Figure 3: As existing LLMs fail to generate long enough output, AgentWrite adopts a plan-then-write pipeline to obtain a sufficient length output with off-the-shelf LLMs.

**Step II: Write**. After obtaining the writing plan from Step I, we call the LLM serially to complete each subtask, generating the writing content section by section. To ensure the coherence of the output, when we call the model to generate the $n$-th section, we also input the previously generated $n - 1$ sections, allowing the model to continue writing the next section based on the existing writing history. Although this serial manner prevents parallel calls to the model to complete multiple sub-tasks simultaneously, and the input length becomes longer, we show in our validation that the overall coherence and quality of the writing obtained this way are far superior to the output generated in parallel. Our prompt is provided in Appendix C.

## 3.2 VALIDATION

We test the generation length and quality of our proposed AgentWrite method on two long-form writing datasets. The first one is LongWrite-Ruler (introduced in Sec 2), and is used to measure exactly how long of an output the method can provide. The second is our constructed LongBench-Write benchmark, which is mainly used to evaluate how well the model-generated content aligns with user instructions in terms of length and writing quality.

**LongBench-Write**. To evaluate the model's performance on a more diverse range of long-form writing instructions, we collect 120 varied user writing prompts, with 60 in Chinese and 60 in English. They are filtered and uniformly sampled from user logs based on predefined categories. To ensure privacy, we manually rewrite each query to remove any user-related sensitive information while maintaining the diversity and representativeness of the dataset. To better assess whether the model's output length meets user requirements, we ensure that *all these instructions include explicit word count requirements*. We divide these instructions into four subsets based on the word count requirements: 0-500 words, 500-2,000 words, 2,000-4,000 words, and over 4,000 words. Additionally, we categorize the instructions into seven types based on the output type: Literature and Creative Writing, Academic and Monograph, Popular Science, Functional Writing, News Report, Community Forum, and Education and Training. We list the number of data in each subset in Table 1.

During evaluation, we adopt two metrics: one for scoring the output length and another for scoring the output quality. We want the model's output length to be as close as possible to the requirements specified in the instructions. Hence, we compute the output length score $S_l$ using a piecewise linear function (where $l$ is the required length, and $l'$ is the actual output length):

$$S_l = \begin{cases} 100 \cdot \max\left(0, 1 - (l'/l - 1)/3\right) & \text{if } l' > l, \\ 100 \cdot \max\left(0, 1 - (l/l' - 1)/2\right) & \text{if } l' \leq l. \end{cases} \quad (1)$$

In other words, when the output length matches the requirement, the score is a perfect 100. The score linearly decays to 0 when the output length is greater than 4 times or less than 1/3 times the requirement. Since outputs that are too short are often more problematic than those that are too long, we set a higher score attenuation coefficient for outputs that are too short.

Table 1: Key statistics of LongBench-Write.

| # Data in each subset | | | |
|---|---|---|---|
| **Language** | | **Output type** | |
| Chinese | 60 | Literature and Creative Writing | 31 |
| English | 60 | Academic and Monograph | 22 |
| **Output length** | | Popular Science | 18 |
| $[0, 500)$ | 26 | Functional Writing | 17 |
| $[500, 2000)$ | 36 | News Report | 13 |
| $[2000, 4000)$ | 31 | Community Forum | 10 |
| $[4000, 20000)$ | 27 | Education and Training | 9 |
| **Average input length** | | | 88 |
| **Average required output length** | | | 2,772 |
| **Median required output length** | | | 1,550 |

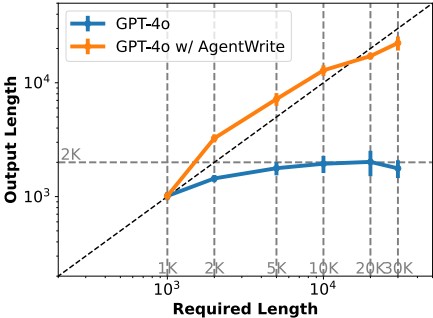

Figure 4: Evaluation on LongWrite-Ruler.

Table 2: Evaluation of AgentWrite strategies on LongBench-Write.

| | **Overall** | | | **[0, 500)** | | **[500, 2k)** | | **[2k, 4k)** | | **[4k, 20k)** | |
|---|---|---|---|---|---|---|---|---|---|---|---|
| | $\bar{S}$ | $S_l$ | $S_q$ | $S_l$ | $S_q$ | $S_l$ | $S_q$ | $S_l$ | $S_q$ | $S_l$ | $S_q$ |
| GPT-4o | 78.6 | 65.3 | 91.8 | 91.0 | 94.6 | 91.4 | 93.6 | 65.5 | 93.0 | 5.6 | 85.3 |
| *+AgentWrite* | 89.1 | 86.6 | 91.6 | 91.0 | 94.6 | 91.4 | 93.6 | 77.3 | 90.2 | 86.8 | 87.5 |
| *+Parallel* | 88.5 | 87.2 | 88.9 | 91.0 | 94.6 | 91.4 | 93.6 | 79.2 | 85.6 | 87.3 | 80.9 |

To automatically evaluate the output quality, we use the LLM-as-a-judge (Zheng et al., 2024; Bai et al., 2024b) approach. Specifically, we select the state-of-the-art GPT-4o (OpenAI, 2024a) model as the judge to score the output across six dimensions: Relevance, Accuracy, Coherence, Clarity, Breadth and Depth, and Reading Experience (please refer to the Appendix C for the scoring prompt). To decouple the quality metric from $S_l$ as much as possible, we instruct the judge model in the prompt to score based solely on the quality of the output, without considering its length. We take the average score across six dimensions to obtain the overall score $S_q$ for output quality. We also provide experimental results in Appendix D to support the consistency and reliability of GPT-4o as a judge for output quality. The final score $\bar{S}$ is computed by the mean of $S_l$ and $S_q$.

**Validation results**. We present the output length measurement on LongWrite-Ruler in Figure 4. We find that AgentWrite successfully extends the output length of GPT-4o from a maximum of 2k words to approximately 20k words. Furthermore, we assess both the output quality and the adherence to the required output length on LongBench-Write. Considering that GPT-4o can successfully complete tasks with outputs under 2,000 words in length when evaluating AgentWrite's performance, we only apply AgentWrite on instructions requiring output lengths of 2,000 words or more. We also assess a variant of AgentWrite, denoted as "*+Parallel*", which calls the model in parallel during Step II to generate outputs for each paragraph.

The results on LongBench-Write are shown in Table 2 (A detailed breakdown of the quality score $S_q$ across different quality dimensions can be found in Table 8). After incorporating AgentWrite, GPT-4o can generate content up to 20k words in length. This significantly improves GPT-4o's length following score ($S_l$), especially in the output length range of [4k, 20k) words. Furthermore, examining the quality score ($S_q$), we can see that AgentWrite does not compromise the quality of the output while expanding its length. By comparing quality scores across six dimensions, we find that AgentWrite significantly improves the Breadth and Depth scores (+5%), while slightly decreasing the Coherence and Clarity scores (-2%). Upon examining the output data, we also notice that outputs generated using AgentWrite occasionally contain minor repetitions. For instance, the model might restate content from previous paragraphs, or frequently provide summarization in its output. Moreover, we find that while *+Parallel* slightly improves the model's output length score, it impairs the output quality of AgentWrite, especially in terms of Coherence (-6%). This suggests that it is necessary to provide the model with the previously generated context in Step II of AgentWrite.

# 4 LONGWRITER: TEACHING MODELS TO GENERATE ULTRA-LONG OUTPUT

Now that we have an agent framework that utilizes off-the-shelf LLMs to automatically generate longer outputs, we are curious: *Is it possible to teach this ability of generating ultra-long outputs to LLMs, allowing them to complete long writing tasks within a single output?* With this question in mind, we conduct model training experiments. In the following sections, we will discuss the construction of training data, model training, and experimental results.

## 4.1 DATA CONSTRUCTION

We first select 6,000 user instructions that *require long outputs (over 2,000 words)* from existing datasets. Specifically, we select 3,000 instructions from GLM-4's SFT data (GLM et al., 2024), mostly in Chinese. Additionally, we select 3,000 instructions from WildChat-1M (Zhao et al., 2024) (a public log of user conversations with ChatGPT/GPT-4), primarily in English. For the automatic selection process, we employ GPT-4o (OpenAI, 2024a), utilizing the prompt provided in Appendix C. We further apply rule-based matching to filter out toxic instructions and those intended for data scraping. We manually check the automatically selected instructions and verify that over 95% of them indeed require responses of several thousand words.

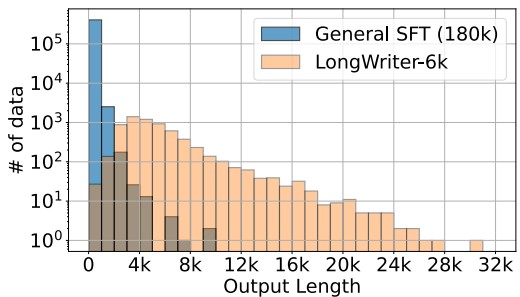

Figure 5: Output length distribution in general SFT dataset and *LongWriter-6k*.

For these 6,000 instructions, we then use the AgentWrite pipeline (introduced in Sec. 3) with GPT-4o to obtain the responses. We further post-process the obtained data, including filtering out outputs that are too short and cases where the model output crashes due to too many planning steps obtained in Step I of AgentWrite. Approximately 0.2% data are filtered out. At the same time, we clean up irrelevant identifiers like "paragraph 1", "paragraph 2", etc., that the model might have added at the beginning of each output section. We call our final obtained long output dataset "*longwriter-6k*".

In model training, to ensure the model's general capabilities, we combine *longwriter-6k* with general SFT data to form the entire training set. In our experiments, we use 180k chat SFT data from GLM-4's SFT data (GLM et al., 2024) as the general SFT data. The output length distribution of the obtained data is displayed in Figure 5. We can see that *LongWriter-6k* effectively supplements the scarcity of general SFT data for output lengths above 2k words, and the output lengths in *LongWriter-6k* are relatively evenly distributed between 2k-10k words.

## 4.2 MODEL TRAINING

**Supervised Fine-tuning**. We conduct training based on two of the latest open-source models, namely GLM-4-9B and Llama-3.1-8B. Both of these are base models and support a context window of up to 128k tokens, making them naturally suitable for training on long outputs. To make the training more efficient, we adopt packing training with loss weighting (Bai et al., 2024a). Our training on the two models results in two models: *LongWriter-9B* (abbr. for LongWriter-GLM-4-9B), and *LongWriter-8B* (abbr. for LongWriter-Llama-3.1-8B).

At the same time, we notice that if we average the loss by sequence, i.e., take the mean of each sequence's average loss within a batch, the contribution of each target token to the loss in long output data would be significantly less than those with shorter outputs. In our experiments, we also find that this leads to suboptimal model performance on tasks with long outputs. Therefore, we choose a loss weighting strategy that averages the loss by token, where the loss is computed as the mean of losses across all target tokens within that batch.

All models are trained using a node with 8xH800 80G GPUs and DeepSpeed+ZeRO3+CPU offloading (Rasley et al., 2020). We use a batch size of 8, a learning rate of 1e-5, and a packing length of 32k. We train the models for 4 epochs, which takes approximately 2,500-3,000 steps.

Table 3: Evaluation results on LongBench-Write. *: Since we utilize GPT-4o to judge the output quality $S_q$, it may bring unfairness when judging itself. The scoring trends on the English subset of LongBench-Write (Table 12) are similar.

| | Overall | | | [0, 500) | | [500, 2k) | | [2k, 4k) | | [4k, 20k) | |
|---|---|---|---|---|---|---|---|---|---|---|---|
| | $\bar{S}$ | $S_l$ | $S_q$ | $S_l$ | $S_q$ | $S_l$ | $S_q$ | $S_l$ | $S_q$ | $S_l$ | $S_q$ |
| *Proprietary models* | | | | | | | | | | | |
| **Claude 3.5 Sonnet** | 80.7 | 73.7 | 87.7 | 87.0 | 92.5 | 93.6 | 90.4 | **81.3** | 86.6 | 26.0 | 80.9 |
| **GPT-4 Turbo** | 67.3 | 47.9 | 86.6 | 92.0 | 90.2 | 81.2 | 90.7 | 12.3 | 85.5 | 0 | 78.7 |
| **GPT-4o mini** | 77.6 | 64.9 | 90.3 | **92.8** | **95.4** | **91.7** | 93.1 | 61.7 | 88.3 | 5.9 | 84.3 |
| **GPT-4o**\* | 78.6 | 65.3 | **91.8** | 91.0 | 94.6 | 91.4 | **93.6** | 65.5 | **93.0** | 5.6 | **85.3** |
| *Open-source models* | | | | | | | | | | | |
| **GLM-4-9B-chat** | 68.3 | 51.0 | 85.5 | 72.8 | 89.9 | 86.6 | 88.5 | 37.9 | 84.8 | 0.2 | 78.7 |
| +AgentWrite | 80.8 | 76.5 | 85.1 | 72.8 | 89.9 | 86.6 | 88.5 | 85.5 | 82.7 | 55.8 | 78.6 |
| **Llama-3.1-8B-Instruct** | 60.3 | 50.0 | 70.6 | 91.0 | 84.0 | 77.9 | 76.6 | 28.1 | 64.5 | 0 | 57.1 |
| +AgentWrite | 71.9 | 73.5 | 70.2 | 91.0 | 84.0 | 77.9 | 76.6 | 72.9 | 63.2 | 51.8 | 56.6 |
| **Llama-3.1-70B-Instruct** | 65.6 | 50.8 | 80.3 | 88.6 | 82.1 | 85.0 | 83.1 | 18.7 | 80.4 | 3.8 | 74.7 |
| +AgentWrite | 80.2 | 82.0 | 78.4 | 88.6 | 82.1 | 85.0 | 83.1 | 88.8 | 75.9 | 63.6 | 71.5 |
| **Mistral-Large-Instruct** | 77.0 | 65.6 | 88.3 | 90.1 | 92.6 | 89.2 | 90.4 | 66.5 | 87.5 | 9.3 | 82.4 |
| **Suri-I-ORPO** | 56.6 | 59.6 | 53.5 | 78.3 | 60.6 | 68.3 | 62.6 | 66.6 | 45.7 | 22.6 | 44.0 |
| *Our trained models* | | | | | | | | | | | |
| **LongWriter-8B** | 79.8 | 77.4 | 82.2 | 80.2 | 82.2 | 74.5 | 82.8 | 78.1 | 83.5 | 77.9 | 79.9 |
| **LongWriter-9B** | 80.5 | 78.6 | 82.3 | 83.9 | 86.2 | 75.6 | 84.8 | 76.0 | 80.2 | 80.3 | 77.3 |
| **LongWriter-9B-DPO** | **84.0** | **82.6** | 85.4 | 82.5 | 88.2 | 81.7 | 86.1 | 76.8 | 85.7 | **90.3** | 81.6 |

**Alignment (DPO)**. To further improve the model's output quality and enhance its ability to follow length constraints in instructions, we perform direct preference optimization (Rafailov et al., 2024) on the supervised fine-tuned LongWriter-9B model. The DPO data comes from GLM-4's chat DPO data (approximately 50k entries). Additionally, we construct 4k pairs of data specifically targeting long-form writing instructions. In particular, for each writing instruction, we sample 4 outputs from LongWriter-9B and score these outputs following the method in Hou et al. (2024). We also combine a length following score as computed in Eq. 1. We then select the highest-scoring output as the positive sample and randomly choose one of the remaining three outputs as the negative sample. The resulting model, *LongWriter-9B-DPO*, is trained for 250 steps on the above data mixture. We follow the recipe in Hou et al. (2024) for DPO training.

It is worth noting that the fairest comparison is between GLM-4-9B-chat and LongWriter-9B-DPO. Both models use GLM-4-9B as the base model and were trained with the same data during the SFT and DPO stages, except for the additional data generated using the AgentWrite method.

## 4.3 EXPERIMENTS

### 4.3.1 MAIN RESULTS

We evaluate 4 proprietary models and 5 open-source models on LongBench-Write (model details listed in Table 5), along with our trained LongWriter models. To the best of our knowledge, Suri-I-ORPO (Pham et al., 2024) is the only prior model that is also aligned for long-form text generation. It is trained based on Mistral-7B-Instruct-v0.2 (Jiang et al., 2023) using LoRA (Hu et al., 2021). We also evaluate open-source versions of AgentWrite, including AgentWrite + GLM-4-9B-chat/Llama-3.1-8B-Instruct/Llama-3.1-70B-Instruct. Consistent with the evaluation setup on LongWrite-Ruler, we set the output temperature to 0.5 and configure the model's generation `max_tokens` parameter to the maximum allowed by its API call. For open-source models, we set it to 32,768. The main results are shown in Table 3. We also report the average and median response length in Table 13. Figure 6 plots the model response length w.r.t. the required length on the 120 instructions in LongBench-Write. Our findings are as follows.

1. **Most previous models are unable to meet the length requirement of over 2,000 words, while LongWriter models consistently provide longer and richer responses to such prompts**. Observing the output length score $S_l$ for prompts in each required length range, we find that previous

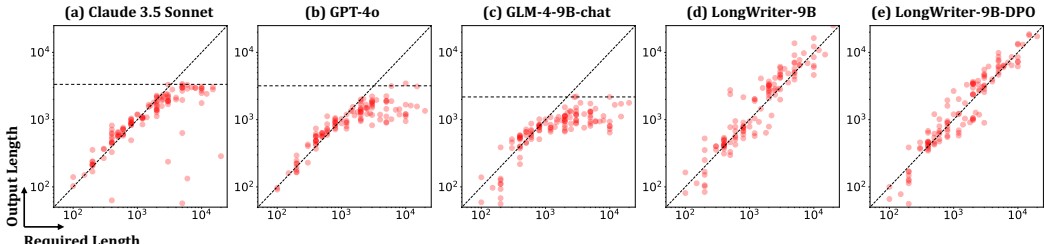

Figure 6: Model response length w.r.t. instruction required length on LongBench-Write.

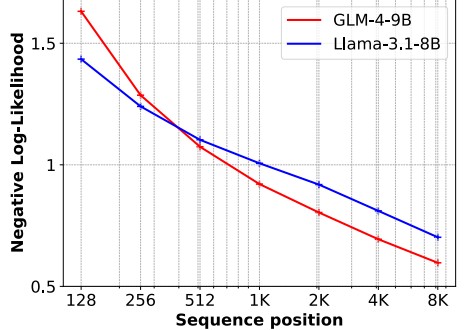

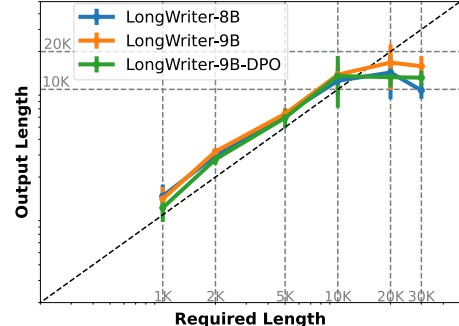

Figure 7: Cumulative average NLL loss of GLM-4-9B and Llama-3.1-8B at different positions of LongWriter models' outputs.

Figure 8: LongWrite-Ruler test results of Long-Writer models, showing their maximum generation lengths between 10k-20k words.

models generally perform poorly (scoring below 70) on prompts in the [2k, 4k) range, with only Claude 3.5 Sonnet achieving a decent score. For prompts in the [4k, 20k) range, almost all previous models are completely unable to reach the target output length, even scoring 0 (meaning all output lengths are less than 1/3 of the required length). By adding training data from LongWriter-6k, our trained model can effectively reach the required output length while maintaining good quality, as suggested by the $S_l$ and $S_q$ on [2k, 20k) range and the scatter plots in Figure 6. We incorporate more careful analysis on the influence of our LongWriter method to the output quality in Appendix E.

We also find that applying the AgentWrite method for open-source models can effectively increase the output length, making it closer to the length requirements of user instructions (higher $S_l$). However, in the $[4k, 20k)$ range, the $S_l$ score is still not high enough, indicating that the output length limit of open-sourced models using the AgentWrite method remains insufficient. Meanwhile, the output quality obtained using the AgentWrite approach on these models is slightly lower compared to direct output (slightly lower $S_q$). Overall, LongWriter models demonstrate better ultra-long-form generation capabilities in terms of both length and quality. Additionally, the inference cost is lower–during the "write" phase of AgentWrite, each round of output requires all the history context from previous outputs and requires re-prefilling, leading to significantly higher inference costs.

To further verify that the long outputs generated by the LongWriter model are coherent and logically connected long texts, rather than simply a concatenation of unrelated segments, we utilize the cumulative average negative log-likelihood test of long context LLMs on the model's outputs. This test is commonly used to evaluate the ability of long context LLMs to model long-range dependencies within long texts (Xiong et al., 2024; Reid et al., 2024). Meanwhile, it can be used inversely: leveraging established long context LLMs to detect the presence of long-range dependencies in long texts, thereby filtering for higher-quality long text data (Chen et al., 2024a). In our testing, we use two existing long context models that support 128k context window: GLM-4-9B and Llama-3.1-8B. Figure 7 reports their cumulative average NLL losses at different positions on approximately 100 text samples longer than 8,192 tokens, generated by three LongWriter models. A lower NLL value indicates better prediction. We observe that both models gain significantly better prediction at later positions, suggesting the prevalence of long-range dependency in LongWriter models' outputs.

**2. DPO effectively improves both the model's output quality and its ability to follow length requirements in long generation**. By comparing the scores of LongWriter-9B and LongWriter-9B-DPO, we find that DPO significantly improves both $S_l$ (+4%) and $S_q$ (+3%) scores, and the improvement is consistent across all ranges. This shows that in long generation scenario, DPO still helps to improve the model's output quality and can better align the model's output length with the requested length. The latter conclusion has also been recently observed in Yuan et al. (2024) in shorter generations. We also manually annotate pairwise wins and losses for GPT-4o and three long-writer models on their outputs in LongBench-Write and visualize the results in Figure 9. Specifically, we invite four annotators to rank the preferences of responses in the form of preference orderings (e.g., $a > b > c > d$). We can see that humans prefer the DPO-trained model over LongWriter-9B in 58% of the cases. Moreover, despite having fewer parameters, LongWriter-9B-DPO achieves a tie with GPT-4o.

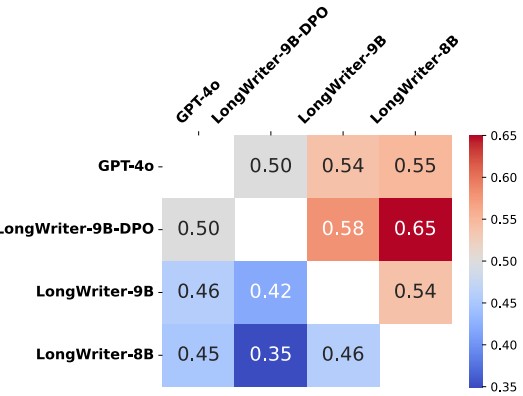

Figure 9: Win-rate heatmap on LongBench-Write.

**3. The output length limit of the LongWriter models is extended to between 10k and 20k words, while more data with long outputs is required to support even longer outputs**. Following the LongWrite-Ruler test in Sec. 2, we also present the LongWrite-Ruler test results of LongWriter models in Figure 8. The results suggest that their maximum generation lengths are between 10k-20k words. The lack of SFT data with longer outputs is likely the primary reason preventing the model from achieving longer output lengths. As seen in Figure 5, there are less than 100 data points with output lengths of 20k words or greater. We believe that constructing longer training SFT data in the future can further push the boundaries of the model's output length limitations, obtaining 100k or even longer output lengths.

### 4.3.2 ABLATION STUDY

We conduct three data ablation experiments on GLM-4-9B, and compare the evaluation results against LongWriter-9B on LongBench-Write. The results are reported in Table 4.

**Ablation on *LongWriter-6k* dataset**. First, we conduct ablation experiments on the *LongWriter-6k* data. As shown in the table, after adding the *LongWriter-6k* dataset, the model (LongWriter-9B) can handle output lengths of 2,000 words and above, as indicated by the output length metric $S_l$. Meanwhile, in terms of $S_q$ (quality), the model trained with the addition of *LongWriter-6k* shows significant improvement (+5%), especially for responses to prompts requiring output lengths in the [2k, 4k] range. We further observed that the improvement in model output quality is mainly in the "Breadth and Depth" dimensions, with an 18% absolute improvement compared to the ablated model. At the same time, as shown in the figure on the right, *LongWriter-6k* data does not bring a bias towards generating longer response.

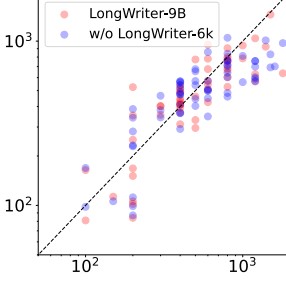

**Ablation on plan-augmented output data**. Previous research has shown that prompting LLMs to externalize their reasoning processes, such as through Chain-of-Thought (Wei et al., 2022) or Tree-of-Thought (Yao et al., 2024), can effectively improve complex task performance. We thus wonder–Would teaching the model to first output the writing plan before generating the writing content be beneficial for long output tasks? To answer this question, we construct a plan-augmented *LongWriter-6k* dataset. Specifically, we concatenate the writing plan obtained through AgentWrite's Step I to the beginning of the writing content, separated by two line breaks, and use the combined text as the output for SFT data. During evaluation, we filter out the writing plan output at the beginning of the model's generation. The results in Table 4 show that the model trained with plan-augmented data slightly improves in output length metric $S_l$ but also decreases in output quality. Overall, teaching the model to first output its reasoning process (writing plan) before generating the

Table 4: Ablation results on LongWriter-9B, evaluated on LongBench-Write: '*-LongWriter-6k data*' is trained with only general SFT data; '*w/ Plan-augmented data*' is trained on general SFT data mixed with plan-augmented *LongWriter-6k* data; *w/ Backtranslation instr.*' is trained on general SFT data mixed with 6k instruction backtranslation data. '+' denotes performance improvement while '-' implies performance degradation ('- -' denotes significant performance degradation).

| | Overall | | | [0, 500) | | [500, 2k) | | [2k, 4k) | | [4k, 20k) | |
|---|---|---|---|---|---|---|---|---|---|---|---|
| | $\bar{S}$ | $S_l$ | $S_q$ | $S_l$ | $S_q$ | $S_l$ | $S_q$ | $S_l$ | $S_q$ | $S_l$ | $S_q$ |
| LongWriter-9B | 80.5 | 78.6 | **82.3** | 83.9 | **86.2** | 75.6 | **84.8** | 76.0 | 80.2 | 80.3 | **77.3** |
| *-LongWriter-6k data* | 62.6 (-) | 48.1 (-) | 77.1 (-) | 83.8 (-) | 85.1 (-) | 77.8 (+) | 79.6 (-) | 25.7 (- -) | 71.9 (-) | 0 (- -) | 71.9 (-) |
| *w/ Plan-augmented data* | **81.4** (+) | **80.9** (+) | 81.8 (-) | **85.9** (+) | 84.0 (-) | **79.4** (+) | 82.3 (-) | **78.2** (+) | **85.2** (+) | **81.4** (+) | 75.0 (-) |
| *w/ Backtranslation instr.* | 60.4 (-) | 44.8 (-) | 70.0 (-) | 80.1 (-) | 81.4 (-) | 77.9 (+) | 77.8 (-) | 18.1 (- -) | 75.0 (-) | 0 (- -) | 69.9 (-) |

writing content does not significantly improve task performance compared to directly outputting the writing content. This might be because the model has already internalized the CoT process when directly learning to generate the writing content (Deng et al., 2024; Yu et al., 2024), thus not relying on explicitly outputting the reasoning process.

**Comparison with instruction backtranslation synthetic data**. We also explore using instruction backtranslation (Li et al., 2024a) to construct long-output SFT data, a method commonly employed in previous LLM long-form generation researches (Wang et al., 2024; Pham et al., 2024). Specifically, we filter text samples (containing both English and Chinese data) with lengths between 2k and 32k words from pretraining datasets and use GLM-4-Long to select those with higher writing quality. We then use GLM-4-Long to generate instructions for these outputs via instruction backtranslation. This results in 6k synthetic data, which are then included in training. As suggested by the result in Table 4, the model trained on backtranslated instruction data fails to meet user requirements for generating longer responses. Its $S_l$ scores do not exceed the model trained only on general SFT data (second row), and the generation quality ($S_q$) is also compromised. We believe that this method is detrimental to the model's learning for two main reasons: 1. Low quality of selected long texts: The long texts used as output sources are not of high quality. Since they originate from pretraining data, many are scraped from web pages, resulting in messy formatting and potential noise. 2. Inconsistency between backtranslated instructions and real user instructions: The backtranslated instructions do not align with the distribution of real user instructions. This prevents the model from learning generalizable capabilities. To further improve the performance of models trained on data constructed using backtranslation, future endeavors may consider collecting higher quality long texts and generating instructions that are more diverse and closer to the distribution of real user instructions.

# 5 RELATED WORK

**Long context LLM**. If we compare an LLM to the human brain, the context window is its working memory. An advanced intelligent being requires a sufficient working memory to accomplish various complex tasks. Similarly, a good LLM needs a long enough context length to replace human on completing these tasks. A line of research has explored how to expand the context window length of LLMs to support long context tasks, allowing the LLM to "see more content and understand longer content". This includes zero-shot extension methods (Han et al., 2023; Xiao et al., 2023; Zhang et al., 2024a; Jin et al., 2024; An et al., 2024), as well as methods that involve fine-tuning the model on longer sequences to achieve a longer memory (Chen et al., 2023a; Peng et al., 2023; Xiong et al., 2024; Chen et al., 2023b; Bai et al., 2024a; Fu et al., 2024). For an intelligent agent with sufficient working memory, they should not only be able to understand longer inputs, but should also possess the ability to produce longer outputs. However, in current long-context LLMs, we find that their maximum output length ($\sim$2,000 words) is far shorter than the maximum context length they can take as input ($>$100,000 words). To bridge this gap, our work studies how to extend the maximum output length of long context LLMs. To the best of our knowledge, Pham et al. (2024) is the only prior research focusing on extending the output length of LLMs, which uses instruction back-translation to construct long output data. As shown by our experiments, this method significantly harms the model's output quality and generalization capability.

**Aligning LLM to follow constraints in instruction**. Since our methodology primarily relies on aligning LLMs to follow user instructions and provide longer, richer outputs, we investigate research on LLM alignment. Prior studies have demonstrated that through alignment training, which involves supervised fine-tuning and reinforcement learning from human feedback (Ouyang et al., 2022; Achiam et al., 2023), LLM can be taught to prioritize privileged instructions (Wallace et al., 2024), follow length constraints (Yuan et al., 2024), and follow multi-constraint instructions (He et al., 2024; Sun et al., 2024; Pham et al., 2024). Our alignment approach specifically tackles the underexplored problem of aligning LLMs to meet user instructions that demand ultra-long outputs.

## 6 Conclusion

In this work, we identify a 2,000-word generation limits for current LLMs, and propose to increase their output window size by adding long-output data during alignment. To automatically construct long-output data, we develop AgentWrite, an agent-based pipeline that uses off-the-shelf LLMs to create extended, coherent outputs. We successfully scale the output window size of current LLMs to 10,000+ words with our constructed *LongWriter-6k*. Extensive ablation studies on the training data demonstrate the effectiveness of our approach. For future work, we suggest the following three directions: 1. Expand the AgentWrite framework to construct data with longer outputs to further extend LLM's output window size. It is also worth exploring the LongWriter method on other tasks, such as generating extensive codebases. 2. Refine the AgentWrite framework to achieve higher quality long-output data, such as adding validation and refinement steps. 3. In our work, we use the vLLM framework (Kwon et al., 2023) for generation: On an 80GB H800 GPU, the LongWriter-9B model takes approximately 55 seconds to produce an output of 10,000 tokens. Longer model outputs bring challenges to inference efficiency. Several methods have been proposed to improve the inference efficiency (Zhang et al., 2024b; Cai et al., 2024; Li et al., 2024b). It is worth investigating how these methods can ensure improved model efficiency without compromising the generation quality.

## Acknowledgement

This work is supported by Beijing Natural Science Foundation (L243006), National Natural Science Foundation of China (62476150), Natural Science Foundation of China (NSFC) 62495063 and 62425601. This work is also supported by Zhipu AI.

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

## A  MODEL CARDS

We list the details of our evaluated models in Table 5.

Table 5: Model cards.

| Model name | Model version | Context window | Max output tokens |
|---|---|---|---|
| Claude 3.5 Sonnet (Anthropic, 2024) | claude-3-5-sonnet-20240620 | 200,000 tokens | 4,096 tokens |
| GPT-4 Turbo (Achiam et al., 2023) | gpt-4-turbo-2024-04-09 | 128,000 tokens | 4,096 tokens |
| GPT-4o mini (OpenAI, 2024b) | gpt-4o-mini-2024-07-18 | 128,000 tokens | 16,384 tokens |
| GPT-4o (OpenAI, 2024a) | gpt-4o-2024-05-13 | 128,000 tokens | 4,096 tokens |
| GLM-4-9B-chat (GLM et al., 2024) | - | 128,000 tokens | - |
| Llama-3.1-8B-Instruct (Dubey et al., 2024) | - | 128,000 tokens | - |
| Llama-3.1-70B-Instruct (Dubey et al., 2024) | - | 128,000 tokens | - |
| Mistral-Large-Instruct (Jiang et al., 2023) | Mistral-Large-Instruct-2407 | 128,000 tokens | - |

## B  LONGWRITE-RULER TEST

We adopt the following 8 seed prompts in our LongWriter-Ruler test:

- Write a $L$-word novel about a teenage heroine who grows up and ends up changing the world
- 写一部讲述一个少女英雄的成长并最终改变世界的$L$字小说
- Write a $L$-word article on the history of the Roman Empire
- 写一篇介绍罗马帝国历史的$L$字文章
- Write a $L$-word paper on the impact of climate change on the global economy
- 写一篇关于气候变化对全球经济影响的$L$字论文
- Write a $L$-word China travel guide
- 写一篇$L$字的中国旅游指南

For each seed prompt, we vary $L \in \{1000, 2000, 5000, 10000, 20000, 30000\}$ and obtain a total of 48 test prompts.

## C  MODEL PROMPTS

**Prompt for AgentWrite step I**.

> I need you to help me break down the following long-form writing instruction into multiple subtasks. Each subtask will guide the writing of one paragraph in the essay, and should include the main points and word count requirements for that paragraph.
> The writing instruction is as follows:
> {*User Instruction*}
> Please break it down in the following format, with each subtask taking up one line:
> Paragraph 1 - Main Point: [Describe the main point of the paragraph, in detail] - Word Count: [Word count requirement, e.g., 400 words]
> Paragraph 2 - Main Point: [Describe the main point of the paragraph, in detail] - Word Count: [word count requirement, e.g. 1000 words].
> ...
> Make sure that each subtask is clear and specific, and that all subtasks cover the entire content of the writing instruction. Do not split the subtasks too finely; each subtask's paragraph should be no less than 200 words and no more than 1000 words. Do not output any other content.

We set the length of each paragraph to be between 200 and 1000 words because: 1. Too short paragraphs ($<$ 200 words) result in an excessive number of paragraphs, making the output overly fragmented. 2. Too long paragraphs ($>$ 1000 words) often lead to GPT-4o generating paragraphs that fall short of the required word count.

**Prompt for AgentWrite step II**.

You are an excellent writing assistant. I will give you an original writing instruction and my planned writing steps. I will also provide you with the text I have already written. Please help me continue writing the next paragraph based on the writing instruction, writing steps, and the already written text.
Writing instruction:
{*User Instruction*}
Writing steps:
{*The writing plan generated in Step I*}
Already written text:
{*Previous generated (n-1) paragraphs*}
Please integrate the original writing instruction, writing steps, and the already written text, and now continue writing {*The plan for the n-th paragraph, i.e., the n-th line in the writing plan*} for me. If needed, you can add a small subtitle at the beginning. Remember to only output the paragraph you write, without repeating the already written text.

**Scoring prompts for quality assessment**.

You are an expert in evaluating text quality. Please evaluate the quality of an AI assistant's response to a user's writing request. Be as strict as possible.
You need to evaluate across the following six dimensions, with scores ranging from 1 to 5. The scoring criteria from 5 to 1 for each dimension are as follows:
1. Relevance: From content highly relevant and fully applicable to the user's request to completely irrelevant or inapplicable.
2. Accuracy: From content completely accurate with no factual errors or misleading information to content with numerous errors and highly misleading.
3. Coherence: From clear structure with smooth logical connections to disorganized structure with no coherence.
4. Clarity: From clear language, rich in detail, and easy to understand to confusing expression with minimal details.
5. Breadth and Depth: From both broad and deep content with a lot of information to seriously lacking breadth and depth with minimal information.
6. Reading Experience: From excellent reading experience, engaging and easy to understand content to very poor reading experience, boring and hard to understand content.
Please evaluate the quality of the following response to a user's request according to the above requirements.
⟨User Request⟩
{User request}
⟨/User Request⟩
⟨Response⟩
{Model response}
⟨/Response⟩
Please evaluate the quality of the response. You must first provide a brief analysis of its quality, then give a comprehensive analysis with scores for each dimension. The output must strictly follow the JSON format: {"Analysis": ..., "Relevance": ..., "Accuracy": ..., "Coherence": ..., "Clarity": ..., "Breadth and Depth": ..., "Reading Experience": ...}. You do not need to consider whether the response meets the user's length requirements in your evaluation. Ensure that only one integer between 1 and 5 is output for each dimension score.

**Prompt for selecting user requests that require 2,000+ word response**.

You will receive an instruction from a user to an AI assistant, please determine whether the instruction requires the AI assistant to write an article, and the length of the article is more than 2,000 words in English (or 2,000 characters in Chinese). If the instruction does not mention the word requirement, please determine whether the user's intention of the response length is more than 2,000 words.
Instruction: {User instruction}
Please judge whether the instruction requires the AI assistant to write an article with more than 2000 words. If yes, please reply "yes", otherwise reply "no", and do not output any other content.

# D    Reliability Analysis of Automated Evaluation

While GPT-4o serves as a primary evaluator, it has shown high consistency in long-text evaluations. To support this, we report the variance of the $S_q$ scores given by GPT-4o in the test results of Table 3 (based on three evaluation runs).

Table 6: Variation of $S_q$ scores over three evaluation runs on LongBench-Write.

| | | | | | |
|---|---|---|---|---|---|
| Claude 3.5 Sonnet | $87.7 \pm 0.5$ | GPT-4 Turbo | $86.6 \pm 0.4$ | GPT-4o mini | $90.3 \pm 0.3$ |
| GPT-4o | $91.8 \pm 0.5$ | GLM-4-9B-chat | $85.5 \pm 0.4$ | Llama-3.1-8B-Instruct | $70.6 \pm 0.3$ |
| Llama-3.1-70B-Instruct | $80.3 \pm 0.3$ | Mistral-Large-Instruct | $88.3 \pm 0.4$ | Suri-I-ORPO | $53.5 \pm 0.5$ |
| LongWriter-8B | $82.2 \pm 0.4$ | LongWriter-9B | $82.3 \pm 0.3$ | LongWriter-9B-DPO | $85.4 \pm 0.3$ |

Moreover, to verify the consistency between GPT-4o and human evaluations of long-output quality, we ask four annotators to independently score the quality of four sets of outputs for 120 prompts from LongBench-Write (resulting in a total of 480 scores). We then calculate the correlation between the human-assigned scores and the GPT-4o scores (*GPT-4o*) as well as the inter-annotator correlations (*Human*).

Table 7: Correlation between human and GPT-4o quality scores.

| | GPT-4o | Human |
|---|---|---|
| Spearman ($\rho$) | 0.51 | 0.55 |
| Kendall ($\tau$) | 0.45 | 0.48 |

Since evaluating writing quality is a relatively subjective task, the table shows that the correlation between human annotators is not very high. Interestingly, the correlation between GPT-4o and human evaluations is close to the inter-annotator correlation, suggesting that GPT-4o can serve as a reliable proxy for automated output quality assessment.

# E    More Evaluation Results

## E.1    More analysis on the output quality of AgentWrite

Table 8: Quality assessment of AgentWrite strategies on LongBench-Write.

| | $S_q$ | Relevance | Accuracy | Coherence | Clarity | Breadth and Depth | Reading Experience |
|---|---|---|---|---|---|---|---|
| GPT-4o | 91.8 | 99.2 | 97.9 | 95.2 | 93.8 | 78.1 | 86.7 |
| +*AgentWrite* | 91.5 | 99.2 | 98.1 | 93.3 | 89.6 | 83.1 | 85.8 |
| +*Parallel* | 88.8 | 97.7 | 95.6 | 88.5 | 86.9 | 80.6 | 83.3 |

To further assess the AgentWrite quality, we manually assess the quality of long responses generated by GPT-4o with or without AgentWrite for 58 queries in LongBench-Write that required responses of 2,000 words or more. Based on the quality of the generated responses, we categorize them into three groups: "High Quality" (meets user requirements, fluent, and clearly expressive), "Good Quality" (generally meets requirements but has minor issues in formatting or expression), and "Poor Quality" (fails to meet requirements or has significant issues with formatting or expression). Note that we only consider the quality of the output content, regardless of whether the length meets the requirement.

Our human evaluation (Table 9) reveals that GPT-4o often produces only an outline for queries requiring ultra-long responses and generates overly general content for professional formats like papers and reports. This lead to a noticeable number of "poor quality" cases during our manual checks. However, after incorporating AgentWrite, such "poor quality" cases are significantly reduced. That

Table 9: Human Eval to assess AgentWrite quality.

|  | #High Quality | #Good Quality | #Poor Quality |
|---|---|---|---|
| GPT-4o | 20 | 18 | 20 |
| +AgentWrite | 26 | 27 | 5 |

said, it also introduces issues such as responses appearing somewhat mechanical (e.g., ending almost every paragraph with a summary or outlook) and occasional minor repetition within the responses.

## E.2 More analysis on the output quality of LongWriter model

We provide detailed scores for six dimensions of generation quality ($S_q$) to illustrate the impact of the LongWriter method on the model's generation quality. From Table 10, it can be seen that LongWriter helps enhance the breadth and depth of the model's output, primarily because the model can generate longer and more detailed content. However, the coherence and clarity of the output are negatively affected, as the model tends to exhibit awkward transitions and occasional pattern repetition when producing longer content. A similar fluctuation in quality is also demonstrated in Table 8: compared to the outputs generated directly by GPT-4o, the long outputs obtained using the AgentWrite method improve breadth and depth but slightly compromise the coherence and clarity of the output. As a result, the LongWriter model trained on such data exhibits similar quality effects when producing long outputs.

Table 10: Quality assessment of LongWriter on LongBench-Write across dimensions.

| Model | $S_q$ | Relevance | Accuracy | Coherence | Clarity | Breadth and Depth | Reading Experience |
|---|---|---|---|---|---|---|---|
| GLM-4-9B-chat | 85.5 | 98.1 | 94.4 | 90.6 | 86.3 | 65.4 | 78.4 |
| LongWriter-9B-DPO | 85.4 | 98.5 (+) | 95.0 (+) | 87.5 (-) | 83.1 (-) | 72.7 (++) | 75.8 (-) |

Table 11: Quality assessment of LongWriter on LongBench-Write across output types.

| Model | LCW | AM | PS | FW | NR | CF | ET |
|---|---|---|---|---|---|---|---|
| GLM-4-9B-chat | 82.6 | 83.0 | 87.5 | 86.8 | 91.3 | 85.6 | 86.6 |
| LongWriter-9B-DPO | 76.1 (-) | 88.1 (+) | 90.1 (+) | 88.3 (+) | 86.9 (-) | 89.8 (+) | 89.0 (+) |

In Table 11, we also provide the quality score $S_q$ of GLM-4-9B-chat and LongWriter-9B-DPO on LongBench-Write queries across seven categories: Literature and Creative Writing (LCW), Academic and Monograph (AM), Popular Science (PS), Functional Writing (FW), News Report (NR), Community Forum (CF), and Education and Training (ET). We can observe that the LongWriter model shows a decline in quality for narrative or creative writing (LCW, NR), while demonstrating an improvement in quality for technical or formal writing (AM, PS, FW, CF, ET). We speculate that this change in output quality is due to the fact that the data constructed by AgentWrite exhibits higher logical consistency, making it suitable for technical writing. However, the segmented generation of outputs by AgentWrite disrupts the coherence of narrative writing, bringing degradation to creative writing quality.

Therefore, to further enhance output quality, future work could improve the AgentWrite pipeline to enhance the coherence of outputs (especially for creative writing types), mitigating the segmentation between and within paragraphs during the generation process.

## E.3 More ways to derive long outputs

One possible way to harvest long outputs is by setting the probability of the end token (`eos_token`) to zero until reaching the desired length during inference. We test the GLM-4-9B-chat model on LongWrite-Ruler queries, setting both `min_new_tokens` and `max_new_tokens` to match the output length specified in the instructions during inference. Let $x$ denote the model output when

min_new_tokens is not set. With the min_new_tokens constraint, the output becomes $[x; y]$, where $y$ is generated because the probability of the eos_token is set to 0 when $x$'s length does not meet the min_new_tokens requirement, forcing the model to continue. We find that in the model's output, $y$ either repeats $x$ (again and again) or consists of repetitive words (even repetitive emojis), adding no meaningful content. We conclude that simply setting the probability of the eos_token to zero is not an effective way to obtain meaningful long outputs from the model.

Table 12: Evaluation results on *English* samples in LongBench-Write.

| | Overall | | | [0, 500) | | [500, 2k) | | [2k, 4k) | | [4k, 20k) | |
|---|---|---|---|---|---|---|---|---|---|---|---|
| | $\bar{S}$ | $S_l$ | $S_q$ | $S_l$ | $S_q$ | $S_l$ | $S_q$ | $S_l$ | $S_q$ | $S_l$ | $S_q$ |
| *Proprietary models* | | | | | | | | | | | |
| **Claude 3.5 Sonnet** | 81.7 | 75.9 | 87.4 | 84.9 | 89.6 | **93.4** | 90.2 | 82.4 | 87.9 | 28.5 | 79.5 |
| **GPT-4 Turbo** | 69.4 | 54.7 | 84.0 | 94.1 | 88.7 | 79.5 | 87.9 | 3.4 | 83.0 | 0 | 70.5 |
| **GPT-4o mini** | 79.2 | 69.2 | 89.2 | **95.0** | **95.3** | 93.2 | 92.7 | 50.8 | 82.2 | 9.3 | 80.0 |
| **GPT-4o** | 79.4 | 67.8 | **90.9** | 92.1 | 93.1 | 92.2 | **93.5** | 53.0 | **92.8** | 6.2 | 81.2 |
| *Open-source models* | | | | | | | | | | | |
| **GLM-4-9B-chat** | 72.4 | 58.4 | 86.3 | 82.6 | 91.7 | 86.7 | 89.0 | 39.8 | 84.5 | 0 | 77.1 |
| **Llama-3.1-8B-Instruct** | 66.6 | 56.8 | 76.3 | 89.7 | 84.6 | 78.2 | 80.6 | 29.2 | 76.1 | 0 | 57.6 |
| **Llama-3.1-70B-Instruct** | 71.2 | 59.0 | 83.3 | 90.8 | 84.8 | 88.6 | 84.4 | 14.9 | 84.5 | 0 | 78.0 |
| **Mistral-Large-Instruct** | 77.6 | 66.7 | 88.5 | 92.5 | 90.2 | 90.0 | 90.8 | 50.0 | 85.6 | 6.5 | **85.1** |
| **Suri-I-ORPO** | 66.6 | 65.5 | 67.6 | 87.8 | 70.6 | 69.4 | 72.4 | 66.8 | 64.8 | 26.4 | 58.3 |
| *Our trained models* | | | | | | | | | | | |
| **LongWriter-8B** | 83.8 | 82.3 | 85.3 | 88.1 | 86.0 | 74.5 | 86.9 | **89.1** | 88.3 | 80.8 | 79.2 |
| **LongWriter-9B** | 83.3 | 83.0 | 83.5 | 86.5 | 85.8 | 72.8 | 84.8 | 88.8 | 84.1 | 89.6 | 77.4 |
| **LongWriter-9B-DPO** | **84.4** | **85.7** | 83.1 | 86.8 | 83.8 | 80.5 | 86.5 | 85.6 | 83.7 | **93.0** | 75.7 |

Table 13: Generation length (# words) statistic in LongBench-Write.

| | [0, 500) | | [500, 2k) | | [2k, 4k) | | [4k, 20k) | |
|---|---|---|---|---|---|---|---|---|
| | Mean | Median | Mean | Median | Mean | Median | Mean | Median |
| **Required Length** | 294 | 300 | 894 | 800 | 2,477 | 2,400 | 8,000 | 6,000 |
| *Proprietary models* | | | | | | | | |
| **Claude 3.5 Sonnet** | 357 | 342 | 927 | 877 | 1,891 | 1,896 | 2,399 | 2,881 |
| **GPT-4 Turbo** | 291 | 294 | 660 | 626 | 778 | 785 | 907 | 701 |
| **GPT-4o mini** | 331 | 317 | 884 | 848 | 2,218 | 1,455 | 1,631 | 1,519 |
| **GPT-4o** | 358 | 386 | 885 | 868 | 1,515 | 1,499 | 1,549 | 1,399 |
| *Open-source models* | | | | | | | | |
| **GLM-4-9B-chat** | 317 | 375 | 758 | 758 | 1,154 | 1,106 | 1,156 | 1,070 |
| **Llama-3.1-8B-Instruct** | 341 | 330 | 819 | 676 | 1,277 | 1,013 | 959 | 991 |
| **Llama-3.1-70B-Instruct** | 331 | 372 | 709 | 720 | 880 | 892 | 1,427 | 1,194 |
| **Mistral-Large-Instruct** | 321 | 308 | 850 | 788 | 1,626 | 1,576 | 1,685 | 1,652 |
| **Suri-I-ORPO** | 539 | 442 | 956 | 804 | 2,193 | 2,149 | 2,668 | 1,941 |
| *Our trained models* | | | | | | | | |
| **LongWriter-8B** | 356 | 374 | 871 | 600 | 4,373 | 3,315 | 7,630 | 6,835 |
| **LongWriter-9B** | 326 | 381 | 1,112 | 778 | 3,371 | 3,171 | 7,528 | 6,678 |
| **LongWriter-9B-DPO** | 317 | 374 | 1,005 | 800 | 2,972 | 3,055 | 8,598 | 7,186 |

