# OpenReview forum: "LongWriter: Unleashing 10,000+ Word Generation from Long Context LLMs"
_ICLR.cc/2025/Conference — ICLR 2025 Poster_

### Official Review · Reviewer_ucCz · 2024-10-30

**Soundness:** 3
**Presentation:** 3
**Contribution:** 3
**Rating:** 6
**Confidence:** 4

**Summary:**

The SOTA LLMs can support the input lenght up to million tokens. However, their output length is limited to a few thousand, which greatly constrains their application in many areas. This work first investigates the reason of such limitation and found that the main reason is because the SFT training sample lenghts are limited. To address this issue, this work introduces an agentic pipleine, AgentWrite, that can generate long output responses by decomposing long generation tasks into subtasks. By using the data generated by this pipeline to train SFT LLMs, the trained LLMs are able to generate responses over 10,000 words while maintaining output quality.
The main contributaions of this work include the following:
1. Analyze and identify the primary reason limiting the output lenght of LLMs.
2. It proposes an agentic framework to construct SFT samples with long output lengths. It also contributes the LongWriter-6k dataset.
3. The empirical results show that the proposed approach is able to scale the current LLMs output lenght to up to 10,000 words.

**Strengths:**

1. This work studies an interesting question of LLM output length limitation, compared to othere works that focus on input lenght. It investigates and finds the reason why output lenght is limited for current LLMs. Specifically, the output lenght is mainly limited by the output length of the SFT training samples.
2. Besides finding the reason of model response lenght limitation, this work also proposes the AgentWrite approach for generating long response SFT data. LLMs trained with this dataset is able to scale its output lenght. The empirical results also show its efficacy.
3. The experiments involves 4 proprietary and 5 open-source models, which makes the conclusions/results of the experiments more convincing.

**Weaknesses:**

1. This work including experiments focsues on only one task, i.e., writing. However, it's not clear how generalizable this approach is to other areas such as coding, which usualy has long output length, especially for large coding projects. Conducting experiments on more tasks can make the proposed approach more compelling.
2. The related work section doesn't mention if there's any exsiting work on this topic. If yes, they should be used as baseline. If no, it should be mentioned explicitly.

**Questions:**

Please see the Weaknesses section.

---

> ### Author Response · Authors · 2024-11-21
> **Thanks for your valuable review**
>
> Thanks for carefully reading our paper! We address the weaknesses as follows.
>
> - **Weaknesses 1: Generalizable of LongWriter to other tasks**
>
> Thanks for your suggestion! We acknowledge that it is worth exploring the LongWriter method on other tasks, such as generating extensive codebases. In this work, we primarily focus on ultra-long-form writing tasks, and we hope to explore a broader range of ultra-long generation tasks in future work.
>
> - **Weaknesses 2: More elaboration on related works**
>
> To the best of our knowledge, prior to this work, the only research focusing on extending the output length of LLMs was Suri [1]. We have compared our model with Suri in the experiments (Table 3), where the results demonstrate that the LongWriter models have significant advantages over the Suri-I-ORPO model in both output quality and length. Thank you for your suggestion, we will include a discussion of [1] in the related work section.
>
> [1] Pham C M, Sun S, Iyyer M. Suri: Multi-constraint instruction following for long-form text generation[J]. arXiv preprint arXiv:2406.19371, 2024.

---

> > ### Comment · Reviewer_ucCz · 2024-11-27
> >
> > Besides code generation, what else long generation tasks you plan to do in the future work?

---

> ### Author Response · Authors · 2024-11-27
>
> For future work on long generation, in addition to areas like creative writing and long code generation, we believe there are promising directions such as academic paper writing (which requires being more technical and leveraging RAG and external tool assistance), as well as scenarios where both input and output involve lengthy texts, such as long-form translation, error correction, and content integration.
>
> We humbly believe that our LongWriter work has explored a feasible path for these potential directions, including data construction and model training. We also look forward to seeing more future work on long generation empowering more fields.

---

> > ### Comment · Reviewer_ucCz · 2024-11-27
> >
> > Makes sense.

---

### Official Review · Reviewer_uF3h · 2024-11-02

**Soundness:** 2
**Presentation:** 3
**Contribution:** 2
**Rating:** 6
**Confidence:** 4

**Summary:**

The paper addresses the limitation of current long context large language models (LLMs) in generating lengthy outputs despite their capacity to process extensive inputs. The authors identify the primary constraint as the scarcity of long-output examples in supervised fine-tuning (SFT) datasets. To overcome this, they introduce AgentWrite, an agent-based pipeline that decomposes ultra-long generation tasks into subtasks, enabling LLMs to produce coherent long outputs. They construct the LongWriter-6k dataset, containing 6,000 SFT data points with output lengths from 2k to 32k words, and incorporate it into model training, successfully scaling output length to over 10,000 words while maintaining quality. Additionally, they develop LongBench-Write, a benchmark for evaluating ultra-long generation capabilities, demonstrating their 9B parameter model's state-of-the-art performance.

**Strengths:**

1. The paper is written clearly, making it easy to understand.
2. This paper focuses on the very practical issue of model output length limitations and provides a systematic research approach.

**Weaknesses:**

I believe there's room for improvement in the experimental aspect.
1. Is it possible to directly leverage AgentWrite to generate long responses with the target model, e.g., Llama-3.1-8B? Can it meet our requirements for output length and quality? Can the output be used to train the model? I think only having results from GPT4-o is not sufficient to demonstrate the effectiveness of AgentWrite.
2. As shown in Table 3, the performance of LongWriter-9B (w/ and w/o DPO) and LongWriter-8B is worse than the original model in the [0, 500) subset, and the performance of LongWriter-9B is also worse in the [500, 2k) subset. Given that only "over 1%" of user prompts require such lengthy outputs, is it worth sacrificing performance on shorter texts? Furthermore, to me, LONGWRITER seems to be essentially about adjusting the model's generation behaviors with longer data plus using GPT4-o for knowledge distillation. The improvement in long-text generation quality through knowledge distillation seems trivial, and changing the model's generation behaviors in this way does not seem to be a good way to enhance the model's fundamental capabilities, as demonstrated by the declined performance on shorter outputs.

Missing reference: Line 189, "Sec"

**Questions:**

1. If we directly set min_new_tokens = max_new_tokens = target length during generation, how would the quality of output from various models be affected?
2. I have doubts about LongWrite-Ruler: the current prompts provided are too vague. If we provide more detailed prompts, will the model generate longer outputs (also with higher quality)? For now, it feels like an assessment of the model's capability to follow instructions in generating outputs of a certain length. Also, as shown in Table 2, even after training the model with longer texts, the model's output still largely fails to meet the required length, which seems to confirm that the model lacks this instruction-following ability, rather than being incapable of generating the required output.

---

> ### Author Response · Authors · 2024-11-21
> **Thanks for your helpful review**
>
> Thanks for your valuable review. We address the weaknesses as follows.
>
> - **Weaknesses 1: Directly leverage AgentWrite to generate long responses with the target model**
>
> Thanks for your suggestion. **We've incorporated comparison to directly leveraging AgentWrite to open-sourced models, including AgentWrite + GLM-4-9B-chat / Llama-3.1-8B-Instruct / Llama-3.1-70B-Instruct**. Note that we only apply AgentWrite to instructions in LongBench-Write that requires output lengths of 2,000 words or more.
>
> |  | $\bar{S}$ | $S_l$ | $S_q$ | $[2k, 4k)$-$S_l$ | $[2k, 4k)$-$S_q$ | $[4k, 20k)$-$S_l$ | $[4k, 20k)$-$S_q$ |
> | ----------- | ----------- | ----------- | ----------- | ----------- | ----------- | ----------- | ----------- |
> | GLM-4-9B-chat | 68.3 | 51.0 | **85.5** | 37.9 | 84.8 | 0.2 | 78.7 |
> | $\qquad$+AgentWrite | 80.8 | 76.5 | 85.1 | 85.5 | 82.7 | 55.8 | 78.6 |
> | Llama-3.1-8B-Instruct | 60.3 | 50.0 | 70.6 | 28.1 | 64.5 | 0 | 57.1 |
> | $\qquad$+AgentWrite | 71.9 | 73.5 | 70.2 | 72.9 | 63.2 | 51.8 | 56.6 |
> | Llama-3.1-70B-Instruct | 65.6 | 50.8 | 80.3 | 18.7 | 80.4 | 3.8 | 74.7 |
> | $\qquad$+AgentWrite | 80.2 | 82.0 | 78.4 | **88.8** | 75.9 | 63.6 | 71.5 |
> | LongWriter-8B | 79.8 | 77.4 | 82.2 | 78.1 | 83.5 | 77.9 | 79.9 |
> | LongWriter-9B-DPO | **84.0** | **82.6** | 85.4 | 76.8 | **85.7** | **90.3** | **81.6** |
>
> We found that applying the AgentWrite method for open-sourced models can effectively increase the output length, making it closer to the length requirements of user instructions (higher $S_l$). However, in the $[4k, 20k)$ range, the $S_l$ score is still not high enough, indicating that the output length limit of open-sourced models using the AgentWrite method remains insufficient. Meanwhile, the output quality obtained using the AgentWrite approach on these models is slightly lower compared to direct output (slightly lower $S_q$).
> Overall, LongWriter models demonstrate better ultra-long-form generation capabilities in terms of both length and quality. Additionally, the inference cost is lower--during the "write" phase of AgentWrite, each round of output requires all the history context from previous outputs and requires re-prefilling, leading to significantly higher inference costs compared to single-pass output.
>
> - **Weaknesses 2: Declined performance on shorter outputs**
>
> First, we need to clarify that in Table 3, **GLM-4-9B-chat** should be compared with **LongWriter-9B-DPO**, not **LongWriter-9B**, because **GLM-4-9B-chat** underwent DPO training, which significantly enhances the quality of the model's outputs. Compared to GLM-4-9B-chat, LongWriter-9B-DPO shows a 1%-2% decrease in quality score \( S_q \) for queries in the $[0, 500)$ and $[500, 2k)$ ranges, while exhibiting a 1%-2% increase for queries in the $[2k, 4k)$ and $[4k, 20k)$ ranges. We believe that such small score fluctuations do not indicate that our method compromises the model's general capability. Variations in the final results can arise from model training, model inference, and GPT-4o evaluation.
>
> Thanks for pointing out the missing reference, we've fixed it.
>
> ***
>
> Here are our responses to your questions.
>
> **Q1**
>
> Great idea! We tested the GLM-4-9B-chat model on LongWrite-Ruler queries, setting both `min_new_tokens` and `max_new_tokens` to match the output length specified in the instructions during inference. Let $x$ denote the model output when `min_new_tokens` is not set. With the `min_new_tokens` constraint, the output becomes $[x; y]$, where $y$ is generated because the probability of the `eos_token` is set to 0 when $x$'s length does not meet the `min_new_tokens` requirement, forcing the model to continue. We found that in the model's output, $y$ either repeats $x$ (again and again) or consists solely of repetitive words (or even repetitive emojis), adding no meaningful content. We concluded that simply setting the probability of the `eos_token` to zero is not an effective way to obtain meaningful long outputs from the model.
>
> **Q2**
>
> In addition to testing the model's output length limit on LongWrite-Ruler (Sec 2), it is worth noting that we also evaluated the output length limit on 120 real user queries from LongBench-Write. As shown in Figure 6 of the paper, despite the varied prompts, the output length of previous models consistently remained below 3k words, while LongWriter models can generate up to 10k words. Notably, as observed in Figure 6, the LongWriter models are largely capable of following the length requirements specified in the instructions. We kindly ask the reviewer where the observation that "even after training the model with longer texts, the model's output still largely fails to meet the required length" was derived from.

---

> > ### Comment · Reviewer_uF3h · 2024-11-25
> >
> > I would like to thank the authors for the detailed response, especially regarding the model's performance in generating short texts. However, the approach of adjusting the training data distribution still seems to lack some novelty. I will keep my score unchanged. Good luck.

---

### Official Review · Reviewer_gq6Q · 2024-11-03

**Soundness:** 3
**Presentation:** 3
**Contribution:** 3
**Rating:** 6
**Confidence:** 4

**Summary:**

The paper addresses a very crucial gap in LLM research of overcoming limitations regarding long form output token limits. The paper investigates the root cause for this limitation in a systematic manner and also comes up with a clever solution to augment LLM solutions to 2k+ token limits. An elaborate evaluation and scoring strategy - that takes into account both the number of tokens generated and the quality of those tokens - has been described, along with DPO alignment and intuitive ablation studies.

**Strengths:**

1. The paper presents a systematic approach towards investigating the reason behind limitations around long generations in LLMs.
2. The proposed agentic pipeline is pretty intuitive and seems to solve the issue very effectively.
3. Extensive validation checks and comparison of SOTA models against the finetuned models has been provided.
4. Good work with seeing the lift provided by DPO alignment and further ablation studies to strengthen the hypothesis.

**Weaknesses:**

1. Need for Human Eval to assess AgentWrite quality - While the paper proposes AgentWrite for generating long-form content, the validation of output quality is primarily based on automatic metrics using GPT-4o as a judge. More rigorous human evaluation would strengthen the quality assessment.
2. Dependency on proprietary models - The AgentWrite pipeline relies on GPT-4o for generating training data, which makes the approach dependent on proprietary models and potentially difficult to reproduce. Analysis and comparison with an open-sourced version of AgentWrite would be helpful to understand how scalable and adaptable the pipeline is.
3. Need for Plan validation in Step 1 : The "Plan" phase of the pipeline comes up with the various subtasks needed for the instruction. There needs to be a validation step to assess the quality and relevance of the subtasks being generated in the first place. The "quality metric" defined later would not take this into account.
4. More elaboration needed on the controlled experiment on Section 2 : this subsection mentions using GLM-4-9b as the base model and then its further finetuned with a subset of GLM-4’s chat SFT data. Has the model not been fine-tuned on this data already? Wouldn't it just overfit on the specific data subset again and that might be the reason why the model performed better when you add longer output length instances in the sft data (due to iterative finetuning on the same instances)? This would nullify the hypothesis that the model’s output limit is due to insufficient output length in the SFT data.

**Questions:**

1. How did you determine the optimal paragraph length range (200-1000 words) for AgentWrite? Were other ranges tested?
2. Is there a theoretical upper limit to how much the output length can be scaled using your approach?
3. Could you elaborate more on what motivated the choice of using token-level loss averaging instead of sequence-level averaging during training?
4. Does the quality of generation degrade differently for different types of content (e.g., technical vs creative writing)?

---

> ### Author Response · Authors · 2024-11-21
> **Thanks for your constructive review (PART 1)**
>
> Thanks for your constructive review. We address the weaknesses as follows.
>
> - **Weaknesses 1: Human Eval to assess AgentWrite quality**
>
> Thank you for your suggestion. **We manually assessed the quality of long responses generated by GPT-4o with or without AgentWrite** for 58 queries in LongBench-Write that required responses of 2,000 words or more. Based on the quality of the generated responses, we categorized them into three groups: "High Quality" (meets user requirements, fluent, and clearly expressive), "Good Quality" (generally meets requirements but has minor issues in formatting or expression), and "Poor Quality" (fails to meet requirements or has significant issues with formatting or expression). Note that we only consider the quality of the output content, regardless of whether the length meets the requirement.
>
> |  | #High Quality | #Good Quality | #Poor Quality |
> | ----------- | :-----------: | :-----------: | :-----------: |
> | GPT-4o      |  20   |   18   |   20   |
> | $\quad$+AgentWrite      |  26  |   27   |  5   |
>
> In our annotations, we found that GPT-4o often produces only an outline for queries requiring ultra-long responses and generates overly general content for professional formats like papers and reports. This led to a noticeable number of "poor quality" cases during our manual checks. However, after incorporating AgentWrite, such "poor quality" cases were significantly reduced. That said, it also introduced issues such as responses appearing somewhat mechanical (e.g., ending almost every paragraph with a "summary" or "outlook") and occasional minor repetition within the responses.
>
> - **Weaknesses 2: More comparisons with AgentWrite + open-sourced models**
>
> Thanks for your suggestion. **We've incorporated comparison with open-sourced versions of AgentWrite, including AgentWrite + GLM-4-9B-chat / Llama-3.1-8B-Instruct / Llama-3.1-70B-Instruct**. Note that we only apply AgentWrite to instructions in LongBench-Write that requires output lengths of 2,000 words or more.
>
> |  | $\bar{S}$ | $S_l$ | $S_q$ | $[2k, 4k)$-$S_l$ | $[2k, 4k)$-$S_q$ | $[4k, 20k)$-$S_l$ | $[4k, 20k)$-$S_q$ |
> | ----------- | ----------- | ----------- | ----------- | ----------- | ----------- | ----------- | ----------- |
> | GLM-4-9B-chat | 68.3 | 51.0 | **85.5** | 37.9 | 84.8 | 0.2 | 78.7 |
> | $\qquad$+AgentWrite | 80.8 | 76.5 | 85.1 | 85.5 | 82.7 | 55.8 | 78.6 |
> | Llama-3.1-8B-Instruct | 60.3 | 50.0 | 70.6 | 28.1 | 64.5 | 0 | 57.1 |
> | $\qquad$+AgentWrite | 71.9 | 73.5 | 70.2 | 72.9 | 63.2 | 51.8 | 56.6 |
> | Llama-3.1-70B-Instruct | 65.6 | 50.8 | 80.3 | 18.7 | 80.4 | 3.8 | 74.7 |
> | $\qquad$+AgentWrite | 80.2 | 82.0 | 78.4 | **88.8** | 75.9 | 63.6 | 71.5 |
> | LongWriter-8B | 79.8 | 77.4 | 82.2 | 78.1 | 83.5 | 77.9 | 79.9 |
> | LongWriter-9B-DPO | **84.0** | **82.6** | 85.4 | 76.8 | **85.7** | **90.3** | **81.6** |
>
> We found that applying the AgentWrite method for open-sourced models can effectively increase the output length, making it closer to the length requirements of user instructions (higher $S_l$). However, in the $[4k, 20k)$ range, the $S_l$ score is still not high enough, indicating that the output length limit of open-sourced models using the AgentWrite method remains insufficient. Meanwhile, the output quality obtained using the AgentWrite approach on these models is slightly lower compared to direct output (slightly lower $S_q$).
> Overall, LongWriter models demonstrate better ultra-long-form generation capabilities in terms of both length and quality. Additionally, the inference cost is lower--during the "write" phase of AgentWrite, each round of output requires all the history context from previous outputs and requires re-prefilling, leading to significantly higher inference costs compared to single-pass output.
>
> - **Weaknesses 3: Plan validation in AgentWrite Step 1**
>
> Great idea! We are indeed attempting to incorporate validation and refining steps into the AgentWrite pipeline in ongoing works.

---

> ### Author Response · Authors · 2024-11-21
> **Thanks for your constructive review (PART 2)**
>
> - **Weaknesses 4: More elaboration on the controlled experiment in Sec 4.2**
>
> We apologize for missing information in Sec 4.2 regarding the training details of the baseline model and the controlled experiments. To clarify the notations, [GLM-4-9B](https://huggingface.co/THUDM/glm-4-9b) is a pretrained base model, and [GLM-4-9B-chat](https://huggingface.co/THUDM/glm-4-9b-chat) is its aligned version. Specifically, the **GLM-4-9B-chat** model is based on the GLM-4-9B base model, first supervised fine-tuned (SFT) with 180k general SFT data (as shown in Figure 5), followed by training with 50k chat DPO data to obtain the final model. In contrast, in our work, during the SFT phase of the GLM-4-9B base model, we mixed 6k longwriter-6k data into the 180k general SFT data, resulting in the **LongWriter-9B** model. Subsequently, we introduced 4k long-form writing DPO data into the 50k chat DPO data during the DPO training phase, producing the **LongWriter-9B-DPO** model. Therefore, the **GLM-4-9B-chat** model and the **LongWriter-9B-DPO** model form a fair comparison--they both underwent the same SFT + DPO training phases from GLM-4-9B, with the only difference being the inclusion of LongWriter data in LongWriter model's training.
>
> ***
>
> Here are our responses to your questions.
>
> **Q1**
>
> In AgentWrite, we set the length of each paragraph to be between 200 and 1000 words because: 1. Too short paragraphs (<200 words) result in an excessive number of paragraphs, making the output overly fragmented. 2. Too long paragraphs (>1000 words) often lead to GPT-4o generating paragraphs that fall short of the required word count.
>
> **Q2**
>
> We do not have a **theoretical** upper limit for the length of model-generated outputs. From the controlled experiments in Sec 2, we observed that the upper limit of the model's generated length generally aligns with the maximum output length in the SFT training data.
>
> **Q3**
>
> We explained our choice of using token-level loss instead of sequence-level loss in lines 307-312. To elaborate, we found that **when using sequence-level loss, the loss weight assigned to each token in the ultra-long output samples becomes too small, with the majority of the loss weight being allocated to tokens in short sequences**. This makes it difficult for the model to fully learn the ability to generate long outputs.
>
> **Q4**
>
> Here we provide the quality score $S_q$ of GLM-4-9B-chat and LongWriter-9B-DPO on LongBench-Write queries across seven categories (refer to Table 1 in our paper): Literature and Creative Writing (LCW), Academic and Monograph (AM), Popular Science (PS), Functional Writing (FW), News Report (NR), Community Forum (CF), and Education and Training (ET).
>
> |  | LCW | AM | PS | FW | NR | CF | ET |
> | ----------- | ----------- | ----------- | ----------- | ----------- | ----------- | ----------- | ----------- |
> | GLM-4-9B-chat | 82.6 | 83.0 | 87.5 | 86.8 | 91.3 | 85.6 | 86.6 |
> | LongWriter-9B-DPO | 76.1 (-) | 88.1 (+) | 90.1 (+) | 88.3 (+) | 86.9 (-) | 89.8 (+) | 89.0 (+) |
>
> We can observe that **the LongWriter model shows a decline in quality for narrative or creative writing (LCW, NR), while demonstrating an improvement in quality for technical or formal writing (AM, PS, FW, CF, ET)**. We speculate that this change in output quality is due to the fact that the data constructed by AgentWrite exhibits higher logical consistency, making it suitable for technical writing. However, the segmented generation of outputs by AgentWrite disrupts the coherence of narrative writing, bringing degradation to creative writing quality. Future work could improve the AgentWrite pipeline to enhance the coherence of outputs for creative writing types, mitigating the segmentation between and within paragraphs during the generation process.

---

> ### Author Response · Authors · 2024-12-01
> **Looking forward to your feedback**
>
> Thank you again for your helpful reviews. As the end of the discussion period draws near, we would like to ensure that we have adequately addressed all your concerns. If you have any further feedback, please do not hesitate to let us know.

---

### Official Review · Reviewer_wuns · 2024-11-09

**Soundness:** 2
**Presentation:** 3
**Contribution:** 2
**Rating:** 6
**Confidence:** 4

**Summary:**

This paper studies the problem of long-form generation for large language models (LLMs). The authors find correlation between the output length limitation of current LLMs and the scarcity of long-output examples in existing supervised fine-tuning datasets. As a remedy, they generate long-output texts by decomposing ultra-long generation (over 10k words)  tasks into subtasks and prompt LLMs for each subtask, and then fine-tune LLMs with the long output. Preference tuning with DPO is also applied. They measure the required output length and the quality of long-form generation judged by GPT-4 as the main metrics. As a result they can enable a 9B/8B parameter model to generate very long sequences over 10k words.

Essentially, the paper focuses on data augmentation for fine-tuning LLMs to a specific problem, which is long-form generation to tens of thousands of words. The idea is straightforward to mix in training examples of ultra-long sequences from 2k to 32k words so that the model generation does not stop after around 2k words.

**Strengths:**

- The focus of the paper is very clear. The paper presents very logical steps around the core idea of enabling LLMs to generate very long sequences. This makes the paper easy to understand (although I feel there might be too many closely related but different names such as LongWrite-Ruler, LongBench-Write, etc. that are a bit confusing).

- The empirical results of demonstrating current LLMs’ cap at around 2k output words provide interesting insights. And connecting it back to the training data distribution is very reasonable leading to the data augmentation used in the paper.

- The data generation pipeline, AgentWrite, is useful for generating long sequences based on instructions. The experimental results are promising, showcasing mixing in long-output sequences can enable the model to generate longer outputs.

**Weaknesses:**

- The novelty of the paper is somewhat limited. I like the idea of enabling the model to generate ultra-long sequences, but essentially it boils down to adjusting the training data length distribution to be better correlated with the testing scenario. Model behaviors are following what models are being trained on, so data augmentation or adjusting training data distribution is always a basic solution.

- The evaluation of ultra-long generation quality is less satisfactory, as it mainly relies on GPT-4. No variance of the LLM-based evaluation is provided, making it a bit hard to gauge the performance differences. From Table 3, it seems the generation quality under this metric is not improved with LongWriter, and the main improvement is on the model becoming able to generate longer sequences, which is not surprising since that was mixed in the training data (e.g. having training examples with end-of-generation token after 10k words instead of 2k).

- Following the above point on long-form generation quality evaluation, there is not enough details provided for the human evaluation in Figure 9 either. Long-form generation is very hard to evaluate, even for humans for texts over 10k words. How can we guarantee that the GPT-4 and human evaluations are trustworthy?

- There are certain baselines lacking for comprehensively comparing long-form generation, compromising the experimental claims. For example, direct comparisons with AgentWrite + other models, which can be a strong baseline with good quality indicated from results in Table 3. Also, for the NLL loss comparison, no baseline models were compared with.

- Efficiency is another concern, which was also mentioned in the conclusion. The paper could discuss more efficiency details such as computational cost, although this might be an orthogonal direction.

**Questions:**

1. Line 194: “we collect 120 varied user writing prompts”: how did you collect?

2. Line 254: “we see that AgentWrite does not compromise the quality of the output while expanding its length”: the scores do drop when the length is expanding in Table 2 right?

3. In Table 3, it seems the quality score judged by GPT-4 does drop compared to other models such as GPT-4o and GLM-4-9B-chat. The main increase of scores for LongWriter come from respecting the required output length. This makes the argument of both length and quality a bit compromised. If users want to generate long output of good quality, it seems AgentWrite is already good or even better.

4. In the paragraph lines 398-409, for NLL comparison, could you compare with other models to see how they score the long generations to make this metric more justifiable?

5. In the ablation study in section 4.3.2, it mentions in lines 476-480 that “taching the model to first output its reasoning process before generating the writing content does not significantly improve task performance…” As I understand this claim is made solely with the GPT-4 evaluation, could there be a problem with the evaluation metric? In fact, I am wondering if the authors have more comments on how to evaluate ultra-long-form generations reliably.


6. Typo: Line 189: “(Introduced in Sec)”

7. Table 4: the green and red colors are not very friendly for accessibility reasons.

---

> ### Author Response · Authors · 2024-11-21
> **Thanks for your comprehensive review (PART 1)**
>
> Thanks for reading so carefully on our paper. We address the weaknesses as follows.
>
> - **Weaknesses 2: Evaluation consistency of GPT-4o as a judge on LongBench-Write**
>
> While GPT-4o serves as a primary evaluator, it has shown high consistency in long-text evaluations. To support this, we report the variance of the $S_q$ scores given by GPT-4o in the test results of Table 3 (based on three evaluation runs).
>
> | Evaluated Models | $S_q$ |
> | ----------- | ----------- |
> | Claude 3.5 Sonnet      | $87.7\pm0.5$     |
> | GPT-4 Turbo   | $86.6\pm0.4$        |
> | GPT-4o mini   | $90.3 \pm0.3$        |
> | GPT-4o   | $91.8 \pm0.5$        |
> | GLM-4-9B-chat   | $85.5 \pm0.4$        |
> | Llama-3.1-8B-Instruct   | $70.6 \pm0.3$        |
> | Llama-3.1-70B-Instruct    | $80.3 \pm0.3$        |
> | Mistral-Large-Instruct    | $88.3 \pm0.4$        |
> | Suri-I-ORPO    | $53.5 \pm0.5$        |
> | LongWriter-8B    | $82.2 \pm0.4$        |
> | LongWriter-9B    | $82.3 \pm0.3$        |
> | LongWriter-9B-DPO    | $85.4 \pm0.3$        |
>
> The reviewer also questioned whether LongWriter improves generation quality. To address this, we provide detailed scores for six dimensions of generation quality ($S_q$) to illustrate the impact of the LongWriter method on the model's generation quality.
>
> | Model | $S_q$ | Relevance | Accuracy | Coherence | Clarity | Breadth and Depth | Reading Experience |
> | ----------- | ----------- | ----------- | ----------- | ----------- | ----------- | ----------- | ----------- |
> | GLM-4-9B-chat | 85.5 | 98.1 | 94.4 | 90.6 | 86.3 | 65.4 | 78.4 |
> | LongWriter-9B-DPO | 85.4 | 98.5 (+) | 95.0 (+) | 87.5 (-) | 83.1 (-) | 72.7 (++) | 75.8 (-) |
>
> It can be seen that **LongWriter helps enhance the breadth and depth of the model's output**, primarily because the model can generate longer and more detailed content. However, **the coherence and clarity of the output are negatively affected**, as the model tends to exhibit awkward transitions and occasional pattern repetition when producing longer content.
>
> A similar fluctuation in quality is also demonstrated in Table 6 of our paper: compared to the outputs generated directly by GPT-4o, the long outputs obtained using the AgentWrite method improve breadth and depth but slightly compromise the coherence and clarity of the output. As a result, the LongWriter model trained on such data exhibits similar quality effects when producing long outputs.
>
> Therefore, to further enhance output quality, future work could focus on improving the AgentWrite pipeline to ensure that the generated long outputs are more coherent and clear.
>
> - **Weaknesses 3: Evaluation correlation of GPT-4o as a judge on LongBench-Write**
>
> For the human comparison win-rate in Figure 9, we invited four annotators to rank the preferences of responses from four models on 120 prompts from LongBench-Write, in the form of preference orderings (e.g., a > b > c > d).
>
> To verify the consistency between GPT-4o and human evaluations of long-output quality, we asked four annotators to independently score the quality of four sets of outputs for 120 prompts from LongBench-Write (resulting in a total of 480 scores). We then calculated the correlation between the human-assigned scores and the GPT-4o scores (*GPT-4o*) as well as the inter-annotator correlations (*Human*).
>
> |  | GPT-4o | Human |
> | ----------- | ----------- | ----------- |
> | Spearman ($\rho$)      |  0.51   |   0.55   |
> | Kendall ($\tau$)      |  0.45  |   0.48   |
>
> Since evaluating writing quality is a relatively subjective task, the table shows that the correlation between human annotators is not very high. Interestingly, **the correlation between GPT-4o and human evaluations is close to the inter-annotator correlation, suggesting that GPT-4o can serve as a reliable proxy for automated output quality assessment**.

---

> ### Author Response · Authors · 2024-11-21
> **Thanks for your comprehensive review (PART 2)**
>
> - **Weaknesses 4: More comparisons with AgentWrite + other models**
>
> Thanks for your suggestion to add AgentWrite + other models as baselines. **We've added the results of AgentWrite + GLM-4-9B-chat / Llama-3.1-8B-Instruct / Llama-3.1-70B-Instruct to Table 3**. Note that we only apply AgentWrite to instructions in LongBench-Write that requires output lengths of 2,000 words or more.
>
> |  | $\bar{S}$ | $S_l$ | $S_q$ | $[2k, 4k)$-$S_l$ | $[2k, 4k)$-$S_q$ | $[4k, 20k)$-$S_l$ | $[4k, 20k)$-$S_q$ |
> | ----------- | ----------- | ----------- | ----------- | ----------- | ----------- | ----------- | ----------- |
> | GLM-4-9B-chat | 68.3 | 51.0 | **85.5** | 37.9 | 84.8 | 0.2 | 78.7 |
> | $\qquad$+AgentWrite | 80.8 | 76.5 | 85.1 | 85.5 | 82.7 | 55.8 | 78.6 |
> | Llama-3.1-8B-Instruct | 60.3 | 50.0 | 70.6 | 28.1 | 64.5 | 0 | 57.1 |
> | $\qquad$+AgentWrite | 71.9 | 73.5 | 70.2 | 72.9 | 63.2 | 51.8 | 56.6 |
> | Llama-3.1-70B-Instruct | 65.6 | 50.8 | 80.3 | 18.7 | 80.4 | 3.8 | 74.7 |
> | $\qquad$+AgentWrite | 80.2 | 82.0 | 78.4 | **88.8** | 75.9 | 63.6 | 71.5 |
> | LongWriter-8B | 79.8 | 77.4 | 82.2 | 78.1 | 83.5 | 77.9 | 79.9 |
> | LongWriter-9B-DPO | **84.0** | **82.6** | 85.4 | 76.8 | **85.7** | **90.3** | **81.6** |
>
> We found that applying the AgentWrite method for open-sourced models can effectively increase the output length, making it closer to the length requirements of user instructions (higher $S_l$). However, in the $[4k, 20k)$ range, the $S_l$ score is still not high enough, indicating that the output length limit of open-sourced models using the AgentWrite method remains insufficient. Meanwhile, the output quality obtained using the AgentWrite approach on these models is slightly lower compared to direct output (slightly lower $S_q$).
> Overall, LongWriter models demonstrate better ultra-long-form generation capabilities in terms of both length and quality. Additionally, the inference cost is lower--during the "write" phase of AgentWrite, each round of output requires all the history context from previous outputs and requires re-prefilling, leading to significantly higher inference costs compared to single-pass output.
>
> - **Weaknesses 5: Concerns w.r.t. efficiency**
>
> We are glad to share that we use the vLLM framework for generation: **On an 80GB H800 GPU, the LongWriter-9B model takes approximately 55 seconds to produce an output of 10,000 tokens**. Future work could focus on accelerating generation speed through techniques like KV cache compression and sparse attention. Additionally, exploring architecture variants such as Mamba could provide insights into performance and efficiency for ultra-long outputs.

---

> ### Author Response · Authors · 2024-11-21
> **Thanks for your comprehensive review (PART 3)**
>
> Here are our responses to your questions.
>
> **Q1**
>
> Thank you for pointing this out. The 120 test queries in LongBench-Write were filtered and uniformly sampled from user logs based on predefined categories. To ensure privacy, we manually rewrote each query to remove any user-related sensitive information while maintaining the diversity and representativeness of the dataset.
>
> **Q2**
>
> Thanks for your keen observation. From Table 2, we can see that the output quality score $S_q$ decreased slightly from 91.8 to 91.6 after using AgentWrite. This difference falls within the margin of error for the GPT-4o evaluator, so we consider it not statistically significant, meaning the output quality can be regarded as essentially unchanged. In Table 6 of the appendix, we show the changes in scores across different dimensions of output quality. After incorporating AgentWrite, the model's outputs exhibit a significant improvement in Breadth and Depth, while Coherence and Clarity show a decline.
>
> **Q3**
>
> Apologies for not emphasizing this in the paper, but in Table 3, the fairest comparison is between GLM-4-9B-chat and LongWriter-9B-DPO. Both models use GLM-4-9B as the base model and were trained with the same data during the SFT and DPO stages, except for the additional data generated using the LongWriter method. As shown, **the two models achieve nearly identical generation quality scores (85.5 vs. 85.4), while LongWriter-9B-DPO demonstrates a significant advantage in output length**.
> We do not claim that the LongWriter method improves the model's generation quality. The primary goal of this method is to better align the model's output length with user instructions that require ultra-long outputs. Additionally, as shown in the table for Weaknesses 4, the model trained with LongWriter achieves higher scores compared to directly applying AgentWrite to an existing model. It is also worth noting that in AgentWrite, each segment of output requires all previous history, causing the input token count during inference to accumulate in a Fibonacci-like manner. This makes it significantly less efficient than generating the entire output in a single pass.
>
> **Q4**
>
> For the NLL loss illustration in Figure 7, we aim to validate the long-range dependencies in LongWriter's long outputs through the continuous downward trend of NLL with token positions, contrary to a simple concatenation of multiple paragraphs (as simple concatenations would not show a consistent decreasing NLL loss curve). Since all baseline models cannot produce such long outputs (8k+ tokens), there is no baseline for the NLL loss study.
>
> **Q5**
>
> Thanks for your intriguing question on evaluating ultra-long-form generations. As we mentioned in our response to Weaknesses 3, the correlation between GPT-4o and human evaluations is close to the inter-annotator correlation, suggesting that GPT-4o can serve as a reliable proxy for automated output quality assessment. Additionally, in our paper, we adopted pairwise comparison as an evaluation method (Figure 9). We believe these approaches can currently serve as reasonably reliable methods for evaluating ultra-long-form generation. However, we also acknowledge that more robust evaluation methods remain an open area for future research. In our current study, we did not come across any universally effective methods for this purpose.
>
> **Q6**
>
> Thanks for pointing this out, we've fixed the typo.
>
> **Q7**
>
> Thank you for your suggestion. We have updated table 4's color scheme and added + and - signs to more clearly illustrate the changes in performance for the readers.

---

> ### Author Response · Authors · 2024-12-01
> **Looking forward to your feedback**
>
> Thank you again for your helpful reviews. As the end of the discussion period draws near, we would like to ensure that we have adequately addressed all your concerns. If you have any further feedback, please do not hesitate to let us know.

---

> > ### Comment · Reviewer_wuns · 2024-12-03
> >
> > I thank the authors for the detailed response and experimental results. Many of my concerns were addressed, thus I increased the score.

---

> > > ### Author Response · Authors · 2024-12-03
> > >
> > > Glad to hear that! Thanks for your support.

---

### Author Response · Authors · 2024-11-21
**Revision**

We sincerely thank all reviewers and ACs for their valuable time and helpful reviews of our paper. In response, we have made significant revisions to our previous manuscript, enriching it with additional content and detailed discussions according to the reviewers' suggestions. We used four colors—red, blue, green, and orange—to correspond to the revision suggestions from the four reviewers (wuns, gq6Q, uF3h, and ucCz). Key additions include:
- **Line 193-195**: How we collect the queries in LongBench-Write (**Reviewer wuns**).
- **Line 336-340, 408-416**: AgentWrite + other models as baseline methods on LongBench-Write evaluation:  (**Reviewers wuns, gq6Q, and uF3h**).
- **Line 352-355**: Clarification on experimental comparison (**Reviewers wuns and gq6Q**).
- **Line 529-532**: Add related work discussion (**Reviewer ucCz**).
- **Line 804-808**: Justification for the 200-1000 words paragraph length range (**Reviewer gq6Q**).
- **Line 866-894**: Consistency and reliability analysis of GPT-4o as a judge on LongBench-Write (**Reviewer wuns**).
- **Line 907-927**: Human Eval to assess AgentWrite quality (**Reviewer gq6Q**).
- **Line 930-945**: Analysis on the output quality of LongWriter model across dimensions (**Reviewer wuns**).
- **Line 946-964**: Analysis on the output quality of LongWriter model across output types (**Reviewer gq6Q**).
- **Line 967-977**: Deriving long outputs by setting min_new_tokens = max_new_tokens = target length (**Reviewer uF3h**).

We thank the reviewers again for their thorough review. Looking forward to your further feedback!

---

### Author Response · Authors · 2024-11-25
**Looking forward to your feedback**

Dear reviewers,

Thank you again for your helpful reviews. As the end of the discussion period draws near, we would like to ensure that we have adequately addressed all your concerns. If you have any further feedback, please do not hesitate to let us know.

---

### Meta-Review · Area_Chair_o64X · 2024-12-21

**Metareview:**

This paper tackles the critical problem of limited output length in LLMs, despite their ability to process long input contexts. The authors identified the core issue as a lack of long-form output examples in standard SFT datasets. The proposed solution, AgentWrite, uses an agent-based approach to decompose ultra-long generation tasks into subtasks, enabling the creation of training data with outputs ranging from 2k to 32k words (LongWriter-6k dataset). Fine-tuning LLMs with this augmented data, combined with preference tuning, demonstrably extends their output capabilities to over 10,000 words while maintaining output quality. The work includes a new benchmark (LongBench-Write) for evaluating ultra-long generation. Reviewers highlighted the paper's clear identification of the problem, the clever and effective data augmentation strategy, and the strong empirical results.

**Additional Comments On Reviewer Discussion:**

Several key weaknesses were raised by reviewers initially. A primary concern is the limited novelty, with the approach being seen as primarily a data augmentation strategy by adjusting the training data length distribution. The evaluation of long-form generation quality was also criticized for its heavy reliance on GPT-4 as a judge without sufficient validation. The lack of strong baselines and limited task diversity further weaken the experimental claims.

Additionally, concerns were raised about the dependency on the proprietary GPT-4 model in the AgentWrite pipeline, the need for validation of the "Plan" phase within AgentWrite, and potential overfitting issues in the controlled experiment. Finally, some reviewers noted a performance decline on shorter outputs after fine-tuning for long-form generation, questioning the trade-off and the fundamental impact of the approach on model capabilities beyond adjusting generation behavior.

Following the rebuttal, most reviewers expressed satisfaction that their concerns had been adequately addressed.

---

### Decision · Program_Chairs · 2025-01-22

Accept (Poster)